# Genomic evidence for global ocean plankton biogeography shaped by large-scale current systems

Daniel J Richter[1,2†], Romain Watteaux[3,4†], Thomas Vannier[5,6,7†], Jade Leconte[5,6], Paul Frémont[5,6], Gabriel Reygondeau[8,9], Nicolas Maillet[10], Nicolas Henry[1,6], Gaëtan Benoit[11], Ophélie Da Silva[6,12], Tom O Delmont[5,6], Antonio Fernàndez-Guerra[13,14,15], Samir Suweis[16], Romain Narci[17], Cédric Berney[1,6], Damien Eveillard[6,18], Frederick Gavory[5], Lionel Guidi[6,12], Karine Labadie[19], Eric Mahieu[19], Julie Poulain[5,6], Sarah Romac[1,6], Simon Roux[20], Céline Dimier[1,21], Stefanie Kandels[22,23], Marc Picheral[6,12], Sarah Searson[6,12], Tara Oceans Coordinators, Stéphane Pesant[24,25], Jean-Marc Aury[5], Jennifer R Brum[20,26], Claire Lemaitre[11], Eric Pelletier[5,6], Peer Bork[22,27,28], Shinichi Sunagawa[22,29], Fabien Lombard[6,12,30], Lee Karp-Boss[31], Chris Bowler[6,21,21], Matthew B Sullivan[20,32,33,34], Eric Karsenti[6,21,23], Mahendra Mariadassou[17], Ian Probert[1,6], Pierre Peterlongo[11], Patrick Wincker[5,6], Colomban de Vargas[1,6*], Maurizio Ribera d'Alcalà[3*], Daniele Iudicone[3*], Olivier Jaillon[5,6*]

*For correspondence:
vargas@sb-roscoff.fr (CV);
maurizio@szn.it (MRd'A);
iudicone@szn.it (DI);
ojaillon@genoscope.cns.fr (OJ)

†These authors contributed equally to this work

Group author details:
Tara Oceans Coordinators See page 16

[1]Sorbonne Université, CNRS, Station Biologique de Roscoff, UMR7144, ECOMAP, Roscoff, France; [2]Institut de Biologia Evolutiva (CSIC-Universitat Pompeu Fabra), Passeig Marítim de la Barceloneta, Barcelona, Spain; [3]Stazione Zoologica Anton Dohrn, Villa Comunale, Naples, Italy; [4]CEA, DAM, DIF, F-91297, Arpajon Cedex, France; [5]Génomique Métabolique, Genoscope, Institut de Biologie François Jacob, CEA, CNRS, Université Evry, Université Paris-Saclay, Evry, France; [6]Research Federation for the study of Global Ocean systems ecology and evolution, FR2O22/Tara GOSEE, Paris, France; [7]Aix Marseille Univ., Université de Toulon, CNRS, IRD, MIO UM, Marseille, France; [8]Changing Ocean Research Unit, Institute for the Oceans and Fisheries, University of British Columbia. Aquatic Ecosystems Research Lab, Vancouver, Canada; [9]Ecology and Evolutionary Biology, Yale University, New Haven, CT, United States; [10]Institut pasteur, Université Paris Cité, Bioinformatics and Biostatistics Hub, Paris, France; [11]Univ Rennes, CNRS, Inria, IRISA-UMR 6074, Rennes, France; [12]Sorbonne Universités, CNRS, Laboratoire d'Oceanographie de Villefranche, LOV, Villefranche-sur-Mer, France; [13]Lundbeck Foundation GeoGenetics Centre, GLOBE Institute, University of Copenhagen, Copenhagen, Denmark; [14]MARUM, Center for Marine Environmental Sciences, University of Bremen, Bremen, Germany; [15]Max Planck Institute for Marine Microbiology, Bremen, Germany; [16]Dipartimento di Fisica e Astronomia 'G. Galilei' & CNISM, INFN, Università di Padova, Padova, Italy; [17]MaIAGE, INRAE, Université Paris-Saclay, Jouy-en-Josas, France; [18]Nantes Université, Ecole Centrale Nantes, CNRS, LS2N, Nantes, France; [19]Genoscope, Institut de biologie François-Jacob, Commissariat à l'Energie Atomique (CEA), Université Paris-Saclay, Evry, France; [20]Department of Microbiology, The Ohio State University, Columbus, United States; [21]Institut de Biologie de l'Ecole Normale Supérieure (IBENS), Ecole Normale Supérieure, CNRS, INSERM, Université PSL, Paris, France; [22]Structural and Computational Biology, European Molecular Biology Laboratory, Heidelberg, Germany; [23]Directors' Research European Molecular Biology

Laboratory, Heidelberg, Germany; [24]MARUM, Center for Marine Environmental Sciences, University of Bremen, Bremen, Germany; [25]PANGAEA, Data Publisher for Earth and Environmental Science, University of Bremen, Bremen, Germany; [26]Department of Oceanography and Coastal Sciences, Louisiana State University, Baton Rouge, United States; [27]Yonsei Frontier Lab, Yonsei University, Seoul, Republic of Korea; [28]Department of Bioinformatics, Biocenter, University of Würzburg, Würzburg, Germany; [29]Institute of Microbiology, Department of Biology, ETH Zurich, Vladimir-Prelog-Weg, Zurich, Switzerland; [30]Institut Universitaire de France (IUF), Paris, France; [31]School of Marine Sciences, University of Maine, Orono, United States; [32]EMERGE Biology Integration Institute, The Ohio State University, Columbus, United States; [33]Center of Microbiome Science, The Ohio State University, Columbus, United States; [34]Department of Civil, Environmental and Geodetic Engineering, The Ohio State University, Columbus, United States

**Abstract** Biogeographical studies have traditionally focused on readily visible organisms, but recent technological advances are enabling analyses of the large-scale distribution of microscopic organisms, whose biogeographical patterns have long been debated. Here we assessed the global structure of plankton geography and its relation to the biological, chemical, and physical context of the ocean (the 'seascape') by analyzing metagenomes of plankton communities sampled across oceans during the *Tara* Oceans expedition, in light of environmental data and ocean current transport. Using a consistent approach across organismal sizes that provides unprecedented resolution to measure changes in genomic composition between communities, we report a pan-ocean, size-dependent plankton biogeography overlying regional heterogeneity. We found robust evidence for a basin-scale impact of transport by ocean currents on plankton biogeography, and on a characteristic timescale of community dynamics going beyond simple seasonality or life history transitions of plankton.

## Editor's evaluation

Richter and colleagues present an impressive analysis of metagenomic, OTU and imaging data collected from >100 ocean locations worldwide, with the purpose of elucidating the role of large-scale currents on global-scale marine plankton biogeography. The topic is exciting and timely.

## Introduction

Plankton communities are constantly on the move, transported by ocean currents. Transport involves both advection and mixing. While being advected by currents, plankton can be influenced by multiple processes, both physicochemical (fluxes of heat, light,and nutrients *Moore et al., 2013*) and biological (species interactions, life cycles, behavior, and acclimation/adaptation [*Armbrust, 2009*; *Flynn et al., 2015*]), which act across various spatial and temporal scales. In turn, plankton impact seawater physicochemistry while they are being advected (*Moore et al., 2013*). The community composition and biogeochemical properties of a water mass at a given site are also partially dependent on its history of mixing with neighboring water masses during transport. These intertwined processes occurring along transport by currents form the pelagic seascape (*Pittman, 2017*; *Figure 1—figure supplement 1a*). Due to logistical and analytical constraints, previous studies on plankton distribution have tended to be geographically or taxonomically restricted (*Hanson et al., 2012*; *Martiny et al., 2006*; *McGowan and Walker, 1979*; *Reygondeau and Dunn, 2019*; *Roux et al., 2016*), to focus on individual factors such as nutrient or light availability (*Longhurst, 2006*; *Tagliabue et al., 2017*), or have investigated the influence of transport on specific nutrients (*Letscher et al., 2016*) or types of planktonic organisms (*Hellweger et al., 2014*; *Villarino et al., 2018*; *Wilkins et al., 2013*). We set out to test for the first time at genomic resolution the hypotheses that a global-scale plankton biogeography exists and that it is closely linked to transport via large-scale ocean currents. To do this, we

**eLife digest** Oceans are brimming with life invisible to our eyes, a myriad of species of bacteria, viruses and other microscopic organisms essential for the health of the planet. These 'marine plankton' are unable to swim against currents and should therefore be constantly on the move, yet previous studies have suggested that distinct species of plankton may in fact inhabit different oceanic regions. However, proving this theory has been challenging; collecting plankton is logistically difficult, and it is often impossible to distinguish between species simply by examining them under a microscope. However, within the last decade, a research schooner called *Tara* has travelled the globe to gather thousands of plankton samples. At the same time, advances in genomics have made it possible to identify species based only on fragments of their DNA sequence.

To understand the hidden geography of plankton communities in Earth's oceans, Richter et al. pored over DNA from the *Tara* Oceans expedition. This revealed that, despite being unable to resist the flow of water, various planktonic species which live close to the surface manage to occupy distinct, stable provinces shaped by currents. Different sizes of plankton are distributed in different sized provinces, with the smallest organisms tending to inhabit the smallest areas. Comparing DNA similarities and speeds of currents at the ocean surface revealed how these might stretch and mix plankton communities.

Plankton play a critical role in the health of the ocean and the chemical cycles of planet Earth. These results could allow deeper investigation by marine modellers, ecologists, and evolutionary biologists. Meanwhile, work is already underway to investigate how climate change might impact this hidden geography.

integrated metagenomic data from epipelagic samples collected during the *Tara* Oceans expedition (***Karsenti et al., 2011***) with in situ and satellite environmental metadata and large-scale ocean circulation simulations. Our sampling largely focused on open ocean sites located in the main gyres, but also included other areas with distinct oceanographic features, such as coastal upwelling zones and lagoons (***Figure 1—figure supplement 1b***). We chose to study biogeographic patterns along large-scale currents in the principal oceanic gyres, with counterpoints to other oceanographic features in which the influence of ocean transport by the main currents is likely to be relatively weaker, such as upwellings. Our analyses focus on the sunlit (epipelagic) layer of the ocean (subsurface and deep chlorophyll maximum [DCM] samples); at lower depths (the mesopelagic and below), the relationship between plankton community composition and ocean transport may be different than at the surface. The use of DNA as a primary proxy for global plankton diversity has several important advantages over classical morphology-based analyses, notably because methods can be standardized and applied across the entire range of plankton sizes, from viruses through prokaryotes and protists to animals.

## Results

DNA sequence data was obtained from samples collected at 113 worldwide stations during the *Tara* Oceans expedition. Each plankton community sample was sequenced for up to six operational size fractions: one virus-enriched (0–0.22 µm; ***Roux et al., 2016***), one prokaryote-enriched (either 0.22–1.6 or 0.22–3 µm; ***Sunagawa et al., 2015***), and four eukaryote-enriched (0.8–5 µm, 5–20 µm, 20–180 µm, and 180–2000 µm; ***de Vargas et al., 2015***; ***Figure 1—figure supplement 1b***). These size fractions are operational in that each contains the organisms captured between two physical filters of a given size (either filters or nets, depending on size fraction [***Pesant et al., 2015***]). We estimated the average percentage of metagenomic sequence reads in samples from the prokaryote-enriched 0.22–1.6/3 µm size fractions that were of eukaryotic origin to be 12%, and the average percentage of reads in eukaryote-enriched size fractions that were of prokaryote origin as follows: 0.8–5 µm: 39%, 5–20 µm: 23%, 20–180 µm: 3%, 180–2000 µm: 5% (see Materials and methods). The *Tara* Oceans project produced a total of 24.2 terabases of metagenomic sequence reads (Supplementary Table 1). To account for uneven sequencing depth among samples, we analyzed a subset of 11.9 terabases, after testing that this subset accurately represented the complete data set (see Materials and methods). We also analyzed operational taxonomic units (OTUs, representing groups

of genetically related organisms), consisting of previously published viral populations (**Brum et al., 2015**) previously derived bacterial 16S miTAGs (**Sunagawa et al., 2015**), and 738 million 18S V9 ribosomal DNA marker sequences in the eukaryote-enriched size fractions, enlarging a previously described *Tara* Oceans data set (**de Vargas et al., 2015**). We used metagenomic data and OTUs independently to compute pairwise comparisons of plankton community dissimilarity (as proxies for β-diversity). Metagenomic dissimilarity highlighted, at species and subspecies resolution, differences in the genomic identity of organisms between stations. Our metagenomic sampling resulted in pairwise metagenomic dissimilarities that likely represent an overestimate of β-diversity (Appendix 1). However, we applied an identical procedure to compute metagenomic dissimilarity for all size fractions (correlations among fractions ranged Spearman's $\rho$ 0.6–0.9, p≤10$^{-4}$, **Figure 1—figure supplement 2**). The more thoroughly sampled OTU dissimilarity, in contrast, incorporated more numerous rare taxa within the plankton, but at genus or higher-level taxonomic resolution (**de Vargas et al., 2015**). Metagenomic and OTU dissimilarities were correlated for all size fractions (Spearman's $\rho$ 0.53–0.97, p≤10$^{-4}$, **Figure 1—figure supplement 2**), indicating that both proxies, although characterized by different sampling levels and taxonomic resolution, provided coherent and complementary estimates of β-diversity (Appendix 1). We performed subsequent analyses using both measures, which produced consistent results. The taxonomic composition of these *Tara* Oceans samples, not discussed here, is instead presented in a parallel analysis of the spatial dynamics of planktonic eukaryotes, based on the same environmental data and large-scale ocean circulation simulations (**Sommeria-Klein et al., 2021**).

We focus on analyses of metagenomic dissimilarity here, with accompanying results for OTU dissimilarity presented in Supplementary Figures, and validation by comparison to abundance differences among metagenome-assembled genomes (MAGs; **Delmont et al., 2022a**) and to more traditional imaging data presented independently below.

Globally, we observed substantial metagenomic dissimilarities (average pairwise dissimilarity >80%) between sampled stations (including adjacent sites) across all size fractions (**Figure 1—figure supplement 3a**, Appendix 1). The resulting portrait is of a heterogeneous oceanic ecosystem at all scales separating *Tara* Oceans sampling sites (even those separated by only a few kilometers), dominated by a small number of abundant and cosmopolitan taxa, with a much larger number of less abundant taxa found at fewer sampling sites (**Figure 1—figure supplement 3b-e**), corroborating other studies (**de Vargas et al., 2015**).

Overlying this heterogeneity, we found robust evidence for the existence of large-scale biogeographical patterns within all plankton size classes using two complementary analyses of dissimilarity among samples (**Figure 1a**, **Figure 1—figure supplement 4a-f**, **Figure 1—figure supplement 5**, Appendix 2). First, we grouped metagenomic samples within each size fraction into 'genomic provinces' via hierarchical clustering (**Figure 1—figure supplement 6**). Second, we derived colors for each sample based on a principal coordinates analysis (PCoA-RGB; see Materials and methods) in order to visualize transitions in community composition within and between genomic provinces. Genomic provinces were mostly composed of geographically clustered stations (consistent with previous studies documenting patterns in plankton biogeography [**Hanson et al., 2012**; **Martiny et al., 2006**; **McGowan and Walker, 1979**; **Roux et al., 2016**; **Figure 1a**, **Figure 1—figure supplement 4a-f**]). Although the large majority of our samples were located in oceanic gyres, samples located in physically distant zones but with shared environmental conditions, such as oceanic upwellings, also grouped together (e.g. genomic province B6 in the bacterial-enriched size fraction). Genomic provinces of smaller plankton (viruses, bacteria, and eukaryotes <20 μm), with some exceptions (e.g. genomic province B5), tended to be limited to a single ocean basin and to approximately correspond to Longhurst biogeochemical provinces (BGCPs; **Longhurst, 2006**; **Figure 1—figure supplement 4a-d**; Appendix 3). In contrast, provinces of larger plankton (micro- and mesoplankton, >20 μm) spanned multiple basins (**Figure 1—figure supplement 4e-f**, Appendix 4).

These large-scale biogeographical patterns derived from metagenomes were linked to environmental parameters including nutrients and temperature. Seawater surface temperature was significantly different among genomic provinces for all plankton size classes (Kruskal-Wallis test, p<10$^{-5}$), corroborating previous results for prokaryotes (**Sunagawa et al., 2015**), whereas other environmental conditions were significantly different only with respect to specific size classes (**Figure 1—figure supplement 7**). The geography of combined nutrient and temperature variations resembled the biogeography of smaller plankton size classes (**Figure 1a–b**, **Figure 1—figure supplement 4a**-d,h),

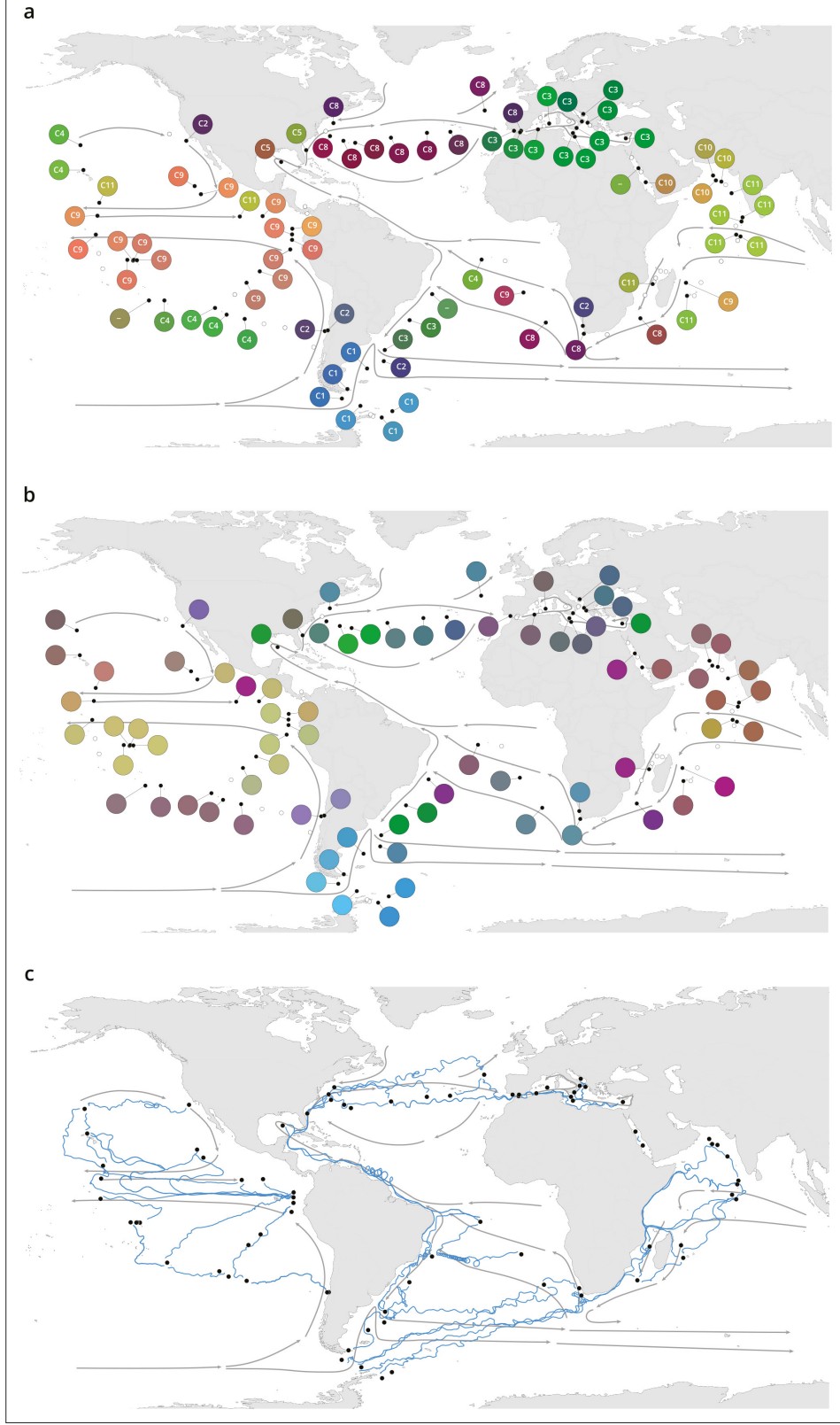

**Figure 1.** Plankton biogeography, environmental variation, and ocean transport among *Tara* Oceans stations. Major currents are represented by solid arrows. (**a**) Genomic provinces of *Tara* Oceans surface samples for the 0.8–5 µm size fraction, each labeled with a letter prefix ('C' represents the 0.8–5 µm size fraction) and a number; samples not assigned to a genomic province are labeled with '-'. Maps of all six size fractions and including deep

*Figure 1 continued on next page*

*Figure 1 continued*

chlorophyll maximum samples are available in *Figure 1—figure supplement 4*. Station colors are derived from an ordination of metagenomic dissimilarities; more dissimilar colors indicate more dissimilar communities (see Materials and methods). (**b**) Stations colored based on an ordination of temperature and the ratio of $NO_3 + NO_2$ to $PO_4$ (replaced by $10^{-6}$ for three stations where the measurement of $PO_4$ was 0) and of $NO_3 + NO_2$ to Fe. Colors do not correspond directly between maps; however, the geographical partitioning among stations is similar between the two maps. (**c**) Simulated trajectories corresponding to the minimum travel time ($T_{min}$) for pairs of stations (black dots) connected by $T_{min}$ <1.5 years. Directionality of trajectories is not represented.

The online version of this article includes the following figure supplement(s) for figure 1:

**Figure supplement 1.** The seascape, plankton transport, and community metagenomic samples of *Tara* Oceans stations.

**Figure supplement 2.** Scatter plots comparing β-diversity estimates from metagenomic, operational taxonomic unit (OTU)-based, and imaging-based dissimilarity.

**Figure supplement 3.** Global dissimilarity and operational taxonomic unit (OTU) occupancy.

**Figure supplement 4.** Genomic provinces in comparison to previous ocean divisions and to metagenome-assembled genome (MAG) abundance variation, and ordination maps of environmental parameters.

**Figure supplement 5.** Biogeography based on an ordination of operational taxonomic unit (OTU) dissimilarity.

**Figure supplement 6.** Hierarchical trees illustrating how samples were partitioned into genomic provinces.

**Figure supplement 7.** Environmental parameters that distinguish genomic provinces.

whereas temperature alone more closely matched the distribution of larger plankton (*Figure 1—figure supplement 4e,f,i*), potentially reflecting different ecological constraints.

Plankton biogeographical patterns suggested a particular role for large-scale surface transport (a core component of the seascape) in the emergence of spatial patterns of plankton community composition (as previously proposed [*Clayton et al., 2013*]), as many genomic provinces were spatially consistent with ocean basin-scale circulation patterns (such as western boundary currents or major subtropical gyres [*Talley et al., 2011*; *Figure 1a*, *Figure 1—figure supplement 4a-f*]). To investigate whether plankton dynamics are related to ocean current timescales, we analyzed community metagenomic composition differences between sampled stations in light of the corresponding transit time, which has previously been suggested as the relevant factor for studying dispersal mechanisms (*Wilkins et al., 2013*). We inferred the characteristic timescale of main transport paths between stations from trajectories computed with the physically well-constrained MITgcm ocean model (see Materials and methods), which takes into account directionalities (*Watson et al., 2010*) and meso- to large-scale circulation, potential dispersal barriers, and mixing effects (*Goetze et al., 2017*; *Mousing et al., 2016*). For this we used the minimum travel time (*Jönsson and Watson, 2016*; $T_{min}$) between pairs of *Tara* stations. These trajectories corresponded to the dominant paths that transport the majority of water volume and its contents (e.g. heat, nutrients, and plankton; *Figure 1c*). For all plankton size classes, community composition differences between stations were significantly correlated to travel time (*Figure 2—figure supplement 1*).

Because the relationships between metagenomic dissimilarities and $T_{min}$ are complex (*Figure 2—figure supplement 1*), global correlations do not necessarily accurately summarize the relationship between communities and currents. To provide more detail on the relationship, we examined cumulative correlation, namely, correlations between community dissimilarity and $T_{min}$ computed for an increasing range of $T_{min}$, which can directly reveal the time window during which plankton dynamics are strongly correlated to ocean current timescales. Cumulative correlation values were maximal for pairs of stations separated by $T_{min}$ <~1.5 years for all size classes, with correlation values (Spearman's $\rho$ 0.45–0.71 depending on size class, p≤$10^{-4}$; *Figure 2a*, *Figure 3—figure supplement 1*) far exceeding those based on previous studies of morphological and/or metabarcode data (*Villarino et al., 2018*) or considering geographic distance rather than travel time (*Louca et al., 2016*). These high correlations between metagenomic dissimilarity and $T_{min}$ for travel times up to 1.5 years, which correspond well with the average time to travel across a basin or gyre (*Lumpkin and Johnson, 2013*), hence reveal measurable plankton community dynamics on time scales far longer than typical plankton growth rates or life cycles. In contrast, no such unimodal pattern was found for correlations between metagenomic

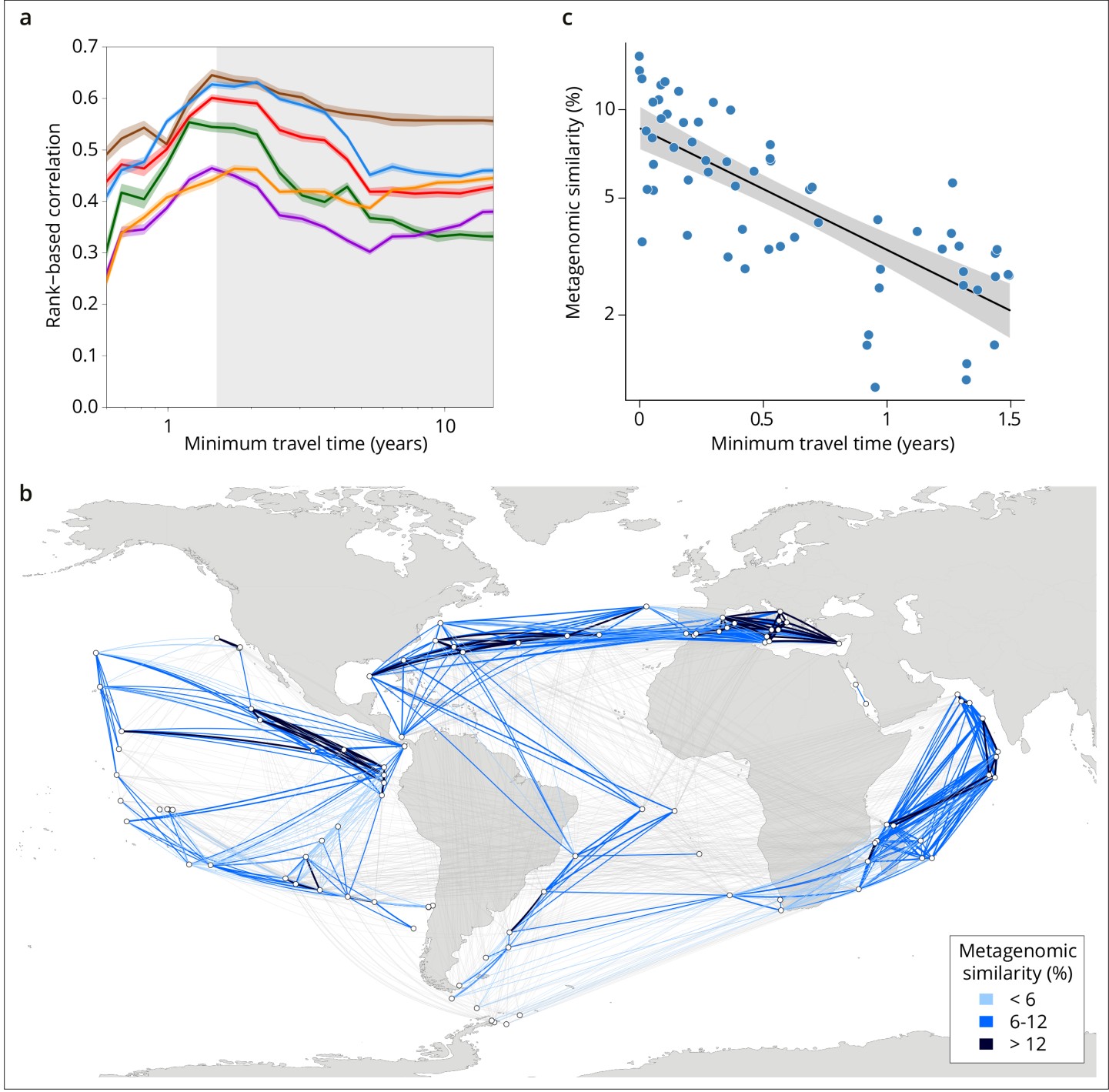

**Figure 2.** Metagenomic dissimilarity and travel time of plankton are maximally correlated up to ~1.5 years. (**a**) Spearman rank-based correlation by size fraction between metagenomic dissimilarity and minimum travel time along ocean currents ($T_{min}$) for pairs of *Tara* Oceans samples separated by a minimum travel time less than the value of $T_{min}$ on the x-axis. Brown line: 0–0.2 μm size fraction, red: 0.22–1.6/3 μm, blue: 0.8–5 μm, green: 5–20 μm, purple: 20–180 μm, orange: 180–2000 μm. Shaded colored areas represent 95% CI. $T_{min}$ >1.5 years is shaded in gray. See plots for operational taxonomic unit (OTU) dissimilarity in ***Figure 3—figure supplement 1***. (**b**) Pairs of *Tara* stations connected by $T_{min}$ <1.5 years in blue/black and >1.5 years in gray. Shading reflects metagenomic similarity from the 0.8–5 μm size fraction. (**c**) The relationship of metagenomic similarity to $T_{min}$ with an exponential fit (black line, gray 95% CI), for pairs of surface samples in the 0.8–5 μm size fraction within the North Atlantic and Mediterranean current system (see map and plots for other size fractions and OTUs in ***Figure 2—figure supplement 2***, and Appendix 1 for a discussion of metagenomic similarity).

The online version of this article includes the following figure supplement(s) for figure 2:

**Figure supplement 1.** Global correlations of dissimilarity with minimum travel time ($T_{min}$).

**Figure supplement 2.** Plankton community composition turnover through the North Atlantic.

dissimilarity and geographic distance (without traversing land; ***Figure 3—figure supplement 1f***).

We compared our analyses of metagenomic data to those based on more traditional zooplankton imaging data collected for the same *Tara* Oceans samples. β-diversity calculated from zooplankton imaging was correlated with metagenomic dissimilarity (Spearman's $\rho$ between 0.32 and 0.60; ***Figure 1—figure supplement 2***), indicating that the two data sources provide concordant measurements of variation in plankton community composition. However, correlations with ocean transport time were far weaker for zooplankton imaging data than for metagenomic data from all organismal size fractions (***Figure 3—figure supplement 1***). We interpret this as being a result of the expected significantly lower resolution in imaging data as compared to metagenomic data (a similar difference of resolution in OTU data versus metagenomic data is discussed in Appendix 1). Finally, we also confirmed our metagenome sequence read comparison-based results by comparing them to β-diversity among sampling sites using a collection of MAGs, which are likely to represent the most abundant genomes, from the 20–180 μm size fraction (the size fraction in which the largest proportion of metagenomic reads were mapped to MAGs, 18.4%; ***Delmont et al., 2022a***). Metagenomic and MAG β-diversity were highly correlated (Spearman's $\rho$ 0.94) and consequently they displayed similar biogeographical patterns (***Figure 1—figure supplement 4e,g***).

Up to ~1.5 years of travel time, the timescale of large-scale transport is therefore the appropriate framework for studying differences in plankton genomic community composition (***Figure 2b***). The fact that simulated transport times and metagenomic dissimilarity were correlated despite a 3-year pan-season sampling campaign, which could be considered to weaken our inference, suggests instead that a large-scale impact of the seascape promotes the existence of a biogeographical structure at a large spatial scale that is resilient to seasonal or other smaller spatiotemporal variations (across all size fractions, genomic provinces consist of stations sampled over an average of 4.7±2.8 different months and 2.7±1.2 different seasons, adjusted for hemisphere). Consistent with our results, seasonal variations have previously been shown to have minor effects on the boundary positions of BGCPs based on satellite data, but not enough to affect the overall pattern of ocean regionalization (***Reygondeau et al., 2013***).

Differences in environmental conditions for pairs of stations also covaried (although less strongly) with transit time for $T_{min}$ < ~1.5 years (***Figure 3***). This indicates that changes in environmental conditions and plankton community composition are concurrent along large-scale oceanic current systems. In our data, beyond ~1.5 years of transport, correlations of $T_{min}$ with metagenomic dissimilarity decreased (***Figures 2a and 3***, ***Figure 3—figure supplement 1a-e***), meaning the signature of transport in the

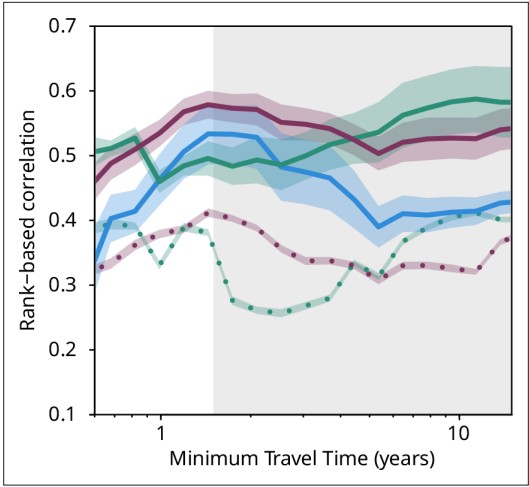

**Figure 3.** Plankton travel time, metagenomic dissimilarity, and environmental differences show different temporal patterns of pairwise correlation. Spearman rank-based correlations between metagenomic dissimilarity and minimum travel time ($T_{min}$, blue), metagenomic dissimilarity and differences in $NO_3 + NO_2$, $PO_4$, and Fe (purple), metagenomic dissimilarity and differences in temperature (turquoise), $T_{min}$ and differences in $NO_3 + NO_2$, $PO_4$, and Fe (purple, dashed), and $T_{min}$ and differences in temperature (turquoise, dashed) for pairs of *Tara* Oceans samples separated by a minimum travel time less than the value of $T_{min}$ on the x-axis. Shaded regions represent SEM. Correlations represent averages across four of six size fractions represented in ***Figure 2a***; the 0–0.2 μm and 5–20 μm size fractions are excluded due to a lack of samples at the global level. Individual size fractions, partial correlations, and correlations with operational taxonomic unit data are in ***Figure 3—figure supplement 1***. Color palette is from microshades (https://github.com/KarstensLab/microshades) (***Dahl et al., 2022***).

The online version of this article includes the following figure supplement(s) for figure 3:

**Figure supplement 1.** Plankton travel time, dissimilarity, environmental distance, and geographic distance show different temporal patterns of pairwise correlation.

timescale of large-scale diversity changes weakened and travel time therefore becomes a less appropriate context to study β-diversity. A similar trend was observed for the correlation between $T_{min}$ and nutrient concentrations, whereas temperature, the gradients of which are mostly dictated by Earth-scale processes that are unaffected by plankton communities, remained well correlated for longer transit times (*Figure 3*).

Together, these analyses suggest the existence in the seascape of biogeochemical continua stretched by currents on the basin scale with predictable, interlinked changes in environmental conditions and plankton community composition (Appendix 5). It has previously been posited that transport could generate continuous transitions between niches based on physical processes (*Lévy et al., 2014*), but it was not anticipated that plankton dynamics would be governed on the time and length scales of main ocean currents. Moreover, beyond ~1.5 years, the correlation of metagenomic dissimilarity with differences in temperature increased while that with differences in nutrients decreased (*Figure 3*, *Figure 3—figure supplement 1a-e*), although both of these correlations with metagenomic dissimilarity remained strong on these time scales. This might be related to distant *Tara* Oceans stations experiencing similar oceanographic phenomena (notably temperature), e.g., upwelling zones (stations 67, 92, and 135; *Figure 1—figure supplement 1*), producing generally similar environmental conditions.

## Discussion

We present the following hypothesis as a potential mechanism for the partitioning of the global ocean into genomic provinces. The relatively large separation (in terms of transport time and season) among sampling stations allowed us to detect the large-scale effects of ocean circulation, which are superimposed on smaller-scale effects such as local patchiness and seasonality (as previously observed *Longhurst, 2006*; *Reygondeau et al., 2013*). Within ocean basins, as the intertwined dynamics of plankton and chemistry continuously occur along transport, smooth variations have emerged due to the periodic recirculation of within-basin currents. This leads to stable, continuous patterns of changes in community structure and nutrient concentrations, and also explains how temporally stable genomic provinces can exist in the face of ocean currents. Among ocean basins, depending on the sensitivity to the environment of a plankton community, higher heterogeneity in environmental conditions across different circulation patterns can disrupt the equilibrium of seascape processes within a given continuum, leading to a global delimitation into distinguishable ecological continua among different large-scale current systems, resulting in the genomic provinces that we detected.

The existence of a size-class-dependent (smaller or larger than 20 μm) structure of plankton geography indicates that the continua that we observe vary among size fractions because of different reactions of organisms within the seascape (i.e. the interplay among organismal biology, nutrients, and local environmental conditions), in agreement with a parallel survey based on taxonomic groups (*Sommeria-Klein et al., 2021*). In the case of the North Atlantic current system (including the Mediterranean Sea), a simple exponential fit of metagenomic dissimilarity along $T_{min}$ for $T_{min} < {\sim}1.5$ years (*Figure 2c*) revealed that the smaller size classes (<20 μm) had a shorter metagenomic turnover time (ca. 1 year) than larger plankton (ca. 2 years; *Figure 2—figure supplement 2*, Appendix 6). At global geographical scales, the genomic provinces of small size classes, which are enriched in phytoplankton (*de Vargas et al., 2015*; *Frémont et al., 2022*; *Sommeria-Klein et al., 2021*; *Sunagawa et al., 2015*), corresponded in our data with differences in environmental parameters such as nutrient levels (*Figure 1b*, *Figure 1—figure supplement 7*) that are often constrained by regional oceanographic processes (*Sarmiento and Gruber, 2006*). On the other hand, genomic provinces of larger plankton, enriched in heterotrophic and symbiotic organisms (*de Vargas et al., 2015*; *Frémont et al., 2022*; *Sommeria-Klein et al., 2021*), were less coupled with geochemical parameters and were more related to global scale gradients and circulation patterns, notably major latitudinal temperature zones (*Martin et al., 2021*) or the separation between Atlantic and Indo-Pacific large-scale surface circulations (*Figure 1—figure supplement 4e,f,i*). These divergent effects were also evident in comparisons of metagenomic dissimilarity with variations in either nutrient concentrations or temperature (*Figure 3—figure supplement 1b*). For smaller plankton, correlations with differences in nutrient concentrations were strongest, at $T_{min}$ up to ~1.5 years, while for larger plankton, correlations were strongest with temperature variations, for $T_{min}$ beyond ~1.5 years. Larger plankton are dominated by eukaryotes, often multicellular, with much longer life cycles, potentially leading to slower community turnover.

Organisms with long life cycles, on the order of several months or years, can be transported through basins spanning multiple biogeochemical niches in which they may encounter strong environmental variability; this trend was also detected in a taxonomy-based analysis accounting for differences in both body size and ecology among groups (*Sommeria-Klein et al., 2021*). As observed here, their biogeography is less affected by nutrient limitation and rather depends on large-scale temperature gradients among basins. This dependence may be linked to the known correlation between body size and organismal metabolic rate (*Ikeda, 1985*). Conversely, variants within populations of organisms with short life cycles have the capacity to increase their relative abundance within restricted ecological niches to which they are adapted. This difference, detectable at genomic resolution, may not be picked up in analyses performed using biological traits with less resolution. These results indicate a significant size-based decoupling within planktonic food webs. For example, large size predators will encounter different prey when transiting through the genomic provinces of small sized organisms (see Appendix 4).

In this study, we provide genomic evidence for an organism size-dependent global-scale open ocean plankton biogeography shaped by currents. Using analysis of standardized metagenomic data together with environmental and physical data, we reveal that, in a background of significant local patchiness, the integration of seascape physical, chemical, and biological processes over time and space produces a quasi-stationary biological partitioning of the oceans that supersedes short-term variability and seasonal cycles, ultimately generating global biogeographical patterns. Although the strong cross-coupling among our metagenomic, environmental, and physical data prevents a systematic disentangling of their various influences, we hypothesize that transport by ocean currents acts essentially as a conveyor on which interacting environmental and biological effects are layered. In this hypothesis, direct effects of currents (such as turbulent diffusivity) on plankton composition are secondary, and instead environmental and biological effects occurring during transport result in the emergence of a global plankton biogeography in the surface ocean. Future studies both on smaller spatiotemporal scales or specific oceanographic features (e.g. coastal regions) and on the global-scale constraints and influences on the seascape itself could lead to a more detailed understanding of plankton dynamics. The ocean is a three-dimensional system in which the primary axis of variation is depth. Our metagenomic data and simulations were limited to the sunlit layer of the ocean and therefore capture one part of seascape dynamics. At greater ocean depths (i.e. in the mesopelagic and below), the relationship between ocean transport and plankton community composition may differ from the one we describe at the surface. In addition, an understanding of plankton biogeography is a key component of future studies on the function of the genes they express; analyses that synergize both characterizations will refine the definition and ecological interpretation of plankton communities within genomic provinces. Overall, our work shows that studies of the dynamics of plankton communities must consider the critical influence of ocean currents in stretching, on the scale of basins, the distribution of both planktonic organisms and the physicochemical nature of the water mass in which they reside. We also demonstrate that the combination of ocean circulation modeling with the use of metagenomic DNA as a tracer of plankton communities provides a resolution above the minimum necessary for assessing the role of transport in community turnover over time and space. The open ocean planktonic ecosystem is fundamentally different in many ways from other major planetary ecosystems, and this study provides a basis to understand and potentially predict the structuring of the ocean ecosystem in a scenario of rapid environmental and current system changes (*Beaugrand et al., 2002*; *Caesar et al., 2018*; *Frémont et al., 2022*).

## Materials and methods
### Sampling, sequencing, and environmental parameters
Sampling, size fractionation, measurement of environmental parameters and associated metadata, DNA extraction, and metagenomic sequencing were conducted as described previously (*Alberti et al., 2017*; *Pesant et al., 2015*). Samples were collected at 113 *Tara* Oceans stations for up to six size fractions (0–0.2, 0.22–1.6/3, 0.8–5, 5–20, 20–180, 180–2000 µm; *Figure 1—figure supplement 1b*; Supplementary Table 1) and two depths (subsurface and DCM). The prokaryote-enriched size fraction was collected either a 0.22–1.6 µm or 0.22–3 µm filter (*Pesant et al., 2015*; *Sunagawa et al.,*

*2015*). For technical reasons, not all size fractions were sequenced for all stations (see Appendix 7 for a summary of why this does not affect our principal conclusions).

We used physicochemical data measured in situ during the *Tara* Oceans expedition (depth of sampling, temperature, chlorophyll *a*, phosphate, nitrate + nitrite concentrations), supplemented with simulated values for iron and ammonium (using the MITgcm Darwin model described below in 'Ocean circulation simulations'), day length, and 8-day averages calculated for photosynthetically active radiation (PAR) in surface waters (AMODIS, https://modis.gsfc.nasa.gov). In order to obtain PAR values at the DCM, we used the following formula (*Morel et al., 2007*):

PAR(Z) = PAR(0) × exp(−k × Z),

x = log(Chl),

log(Z) = 1.524 − 0.426x − 0.0145x^2 + 0.0186x^3,

k = −ln (0.01)/Z,

in which k is the attenuation coefficient, and Z is the depth of the DCM (in meters). Other data, such as silicate and the (nitrate + nitrite)/phosphate ratio, were extracted from the World Ocean Atlas 2013 (WOA13 version 2, https://www.nodc.noaa.gov/OC5/woa13/), by retrieving the annual mean values at the closest available geographical coordinates and depths to *Tara* sampling stations. For temperature and nitrate + nitrite, we calculated seasonality indexes (SI) from monthly WOA13 data. For each sample, the index is the annual variation of the parameter (max - min) at this location divided by the highest variation value among all samples.

A list of samples, metagenomic and metabarcode sequencing information, and associated environmental data are available in Supplementary Tables 1–2.

## Calculation of metagenomic community dissimilarity

Metagenomic community distance between pairs of samples was estimated using whole shotgun metagenomes for all six size fractions. We used a metagenomic comparison method (Simka *Benoit et al., 2016*) that computes standard ecological distances by replacing species counts by counts of DNA sequence k-mers (segments of length *k*). Within Simka, we filtered regions of low complexity with read-shannon-index set to 1.5. K-mers of 31 base pairs (bp) derived from the first 100 million reads sequenced in each sample (or the first 30 million reads for the 0–0.2 µm size fraction) were used to compute a similarity measure between all pairs of samples within each organismal size fraction. Based on a benchmark of Simka, we selected 100 million reads per sample (or 30 million for the 0–0.2 µm fraction) because increasing this number did not produce a qualitatively different set of results, and to ensure that the same number of reads was used in each pairwise comparison within a size fraction. Nearly all samples in our data set had at least 100 million reads (or at least 30 million for the 0–0.2 µm fraction; Supplementary Table 1).

We estimated β-diversity for metagenomic reads with the following equation within Simka:

Metagenomic β-diversity = (b + c)/(2a + b + c)

where a is the number of distinct k-mers shared between two samples, and b and c are the number of distinct k-mers specific to each sample. We represented the distance between each pair of samples on a heatmap using the heatmap.2 function of the R-package (*R Core Team T, 2017*) gplots_2.17.0 (*Warnes et al., 2015*). The dissimilarity matrices we produced for each plankton size fraction (on a scale of 0 = identical to 100 = completely dissimilar) are available as Supplementary Tables 3–8.

## Calculation of OTU-based community dissimilarity

Within the 0–0.2 µm size fraction, we used previously published viral populations (equivalent to OTUs; *Brum et al., 2015*) and viral clusters (analogous to higher taxonomic levels; *Roux et al., 2016*) based on clustering of protein content. For the 0.22–1.6/3 µm size fraction, we used previously derived miTAGs based on metagenomic matches to 16S ribosomal DNA loci and processed them as described (*Sunagawa et al., 2015*). For the four eukaryotic size fractions, we added additional samples to a previously published *Tara* Oceans metabarcoding data set and processed them using the same methods (*de Vargas et al., 2015*; also described at DOI: 10.5281/zenodo.15600).

We calculated OTU-based community dissimilarity for all size fractions as the Jaccard index based on presence/absence data using the vegdist function implemented in vegan 2.4–0 (*Oksanen et al.,*

*2019*) in the software package R. The dissimilarity matrices we produced for each plankton size fraction (on a scale of 0 = identical to 100 = completely dissimilar) are available as Supplementary Tables 9–14.

## Calculating distances of environmental parameters

We calculated Euclidean distances (*Legendre and Legendre, 2012*) for physicochemical parameters. Each were scaled individually to have a mean of 0 and a variance of 1 and thus to contribute equally to the distances. Then the Euclidean distance between two stations *i* and *j* for parameters *P* was computed as follows:

$$ED\left(i,j,P\right) = \sqrt{\sum_{p \in P}\left(x_{ip} - x_{jp}\right)^2}$$

## RGB encoding of environmental positions

We color-coded the position of stations in environmental space for *Figure 1b* and *Figure 1—figure supplement 4h* as follows. First, environmental variables were power-transformed using the Box-Cox transformation to have Gaussian-like distributions to mitigate the effect of outliers and scaled to have zero mean and unit variance. We then performed a principal components analysis (PCA) with the R command prcomp from the package stats 3.2.1 (*R Core Team T, 2017*) on the matrix of transformed environmental variables and kept only the first three principal components. Finally, we rescaled the scores in each component to have unit variance and decorrelated them using the Mahalanobis transformation. Each component was mapped to a color channel (red, green, or blue) and the channels were combined to attribute a single composite color to each station. The components (x, y, z) were mapped to color channel values (r, g, b) between 0 and 255 as r = 128 × (1 + x/max[abs(x)]), and similarly for g and b. This map ensures that the global dispersion is equally distributed across the three components and composite colors span the whole color space.

## Definition of genomic provinces

We used a hierarchical clustering method on the metagenomic pairwise dissimilarities produced by Simka for all surface and DCM samples, and multiscale bootstrap resampling for assessing the uncertainty in hierarchical cluster analysis. We focused on metagenomic dissimilarity due to its higher resolution, and confirmed that the patterns found in metagenomic data were consistent when using OTU data (*Figure 1—figure supplement 5*). We used UPGMA (unweighted pair-group method using arithmetic averages) clustering, as it has been shown to have the best performance to describe clustering of regions for organismal biogeography (*Kreft and Jetz, 2010*). The R-package pvclust_1.3–2 (*Suzuki and Shimodaira, 2006*), with average linkage clustering and 1000 bootstrap replications, was used to construct dendrograms with the approximately unbiased p-value for each cluster (*Figure 1—figure supplement 6*). Because the number of genomic provinces by size fraction was not known apriori, we applied a combination of visualization and statistical methods to compare and determine the consistency within clusters of samples. First, the Silhouette method (*Rousseeuw, 1987*) was used to measure how similar a sample was within its own cluster compared to other clusters using the R package cluster_2.0.1 (*Maechler et al., 2015*). The Silhouette coefficient *s* for a single sample is given as:

$$s = (b - a)/\max(a, b)$$

where *a* is the mean distance between *a* sample and all other points in the same class and *b* is the mean distance between *a* sample and all other points in the next nearest cluster. We used the value of *s*, in addition to bootstrap values, to partition each tree into genomic provinces (see Appendix 2 for further details on statistical validation of genomic provinces). Additionally, we used the Radial Reingold-Tilford Tree representation from the JavaScript library D3.js (https://d3js.org/) (*Bostock et al., 2011*) to visualize sample partitions from the dendrogram. Single samples were not considered as genomic provinces.

In a complementary approach, we performed a PCoA with the R command cmdscale (eig = TRUE, add = TRUE) from the package stats 3.2.1 (*R Core Team T, 2017*) on the matrices of pairwise metagenomic dissimilarities calculated by Simka (or OTU dissimilarity measured with the Jaccard index)

within each size fraction and kept only the first three principal coordinates. We then converted those coordinates to a color using the RGB encoding described above, with one modification: scaling factors $\lambda_r$, $\lambda_g$, and $\lambda_b$ were calculated as the ratios of the second and third eigenvalues to the first (dominant) eigenvalue to ensure that the dispersion of stations along each color channel reproduced the dispersion of the stations along the corresponding principal component (the ratio for the color corresponding to the dominant eigenvalue is 1). The components (x, y, z) were then mapped to color channel values (r, g, b) between 0 and 255 as $r = 128 \times (1 + \lambda_c x/\max[\mathrm{abs}(x)])$, where $\lambda_c$ is the ratio of the eigenvalue of color c to the dominant eigenvalue.

We represented number and PCoA-RGB color of genomic provinces for each sample on a world map (*Figure 1*, *Figure 1—figure supplement 4a-f*) generated with the R packages maps_3.0.0.2 (*Becker et al., 2018*), mapproj 1.2–4 (*McIlroy et al., 2015*), gplots_2.17.0 (*Warnes et al., 2015*), and mapplots_1.5 (*Gerritsen, 2014*). We also plotted phosphate and temperature (*Figure 1—figure supplement 4a-f*) obtained from the *Csiro Atlas of Regional Seas* (CARS2009, http://www.cmar.csiro.au/cars) using the phosphate_cars2009.nc and temprerature_cars2009a.nc files and the R package RNetCDF (*Ridgway et al., 2002*).

## Comparison of genomic provinces to previous ocean divisions

To evaluate the spatial similarity between the clusters obtained in our study for each size fraction and previous biogeographic divisions, we performed an analysis of similarity (ANOSIM, Fathom toolbox, MATLAB). First, we collected coordinates for three spatial divisions at a resolution of $0.5 \times 0.5°$: biomes, BGCPs (*Longhurst, 2006*; *Reygondeau et al., 2013*), and objective global ocean biogeographic provinces (OGOBPs; *Oliver and Irwin, 2008*). Second, we assigned *Tara* Oceans stations to biomes, BGCPs, and OGOBPs based on their GPS coordinates. Third, for each size fraction we performed an ANOSIM with the metagenomic dissimilarity matrix calculated by Simka, using biogeographic clusters (biome, BGCP, and OGOBP) as group membership for each station. Each ANOSIM was bootstrapped 1000 times to evaluate the interval of confidence around the strength of the relationships we detected (*Figure 1—figure supplement 4a-f*).

## Environmental differences among genomic provinces

For each size fraction, we tested which environmental parameters significantly discriminated among genomic provinces (*Figure 1—figure supplement 7*). A total of 12 parameters characterizing each sample, grouped by genomic provinces, were evaluated with a Kruskal-Wallis test within each size fraction with a significance threshold of $p<10^{-5}$. One such parameter, sunshine duration (day length) does not map unambiguously to season, as day lengths coincide in spring and autumn. Ocean biology, chemistry, and stratification often differ between spring and autumn. As such, we provide seasonality indices for temperature and for nitrate + nitrite (described above in the Methods, 'Sampling, sequencing, and environmental parameters'), which represent annual variation in these environmental parameters, and can help interpret the effects of seasonality on genomic provinces. Selected parameters for each size fraction were then used to perform a PCA of the samples using the R package vegan_1.17–11 (*Oksanen et al., 2019*). Samples were plotted with the same PCoA-RGB colors used in the genomic province maps above and each genomic province surrounded by a gray polygon. In analyses where Southern Ocean (including Antarctic) stations were considered independently from other stations, the following were considered Southern Ocean stations: 82, 83, 84, 85, 86, 87, 88, and 89.

## Ocean circulation simulations

We derived travel times from the MITgcm Darwin simulation (*Clayton et al., 2016*) based on an optimized global ocean circulation model from the ECCO2 group (*Menemenlis et al., 2008*). The horizontal resolution of the model was approximately 18 km, with 1,103,735 total ocean cells. We ran the model for six continuous years in order to smooth anomalies that might occur during any single year. We used surface velocity simulation data to compute trajectories of floats originating in ocean cells containing all *Tara* Oceans stations and applied the following stitching procedure to generate a large number of trajectories for each initial position. (The use of surface velocity data implies that Ekman transport also influences trajectories within the simulation.)

First, we precomputed a set of monthly trajectories: for each of the 72 months in the data set, we released floats in every ocean cell of the model grid and simulated transport for 1 month. We used a fourth-order Runge-Kutta method with trilinearly interpolated velocities and a diffusion of 100 m²/s.

Second, following previous studies (*Hellweger et al., 2014*), we stitched together monthly trajectories to create 10,000-year trajectories: for each float released within a 200 km radius of a *Tara* station, we constructed 1000 trajectories, each 10,000 years long. To avoid seasonal effects, we began by selecting a random starting month. We followed the trajectory of a float released within that month to the grid cell containing its end point at the end of the month. Next, we randomly selected a trajectory starting on the following month (e.g. February would follow January) from that grid cell and repeated until reaching a 10,000-year trajectory.

We searched the resulting 50.8 million trajectories for those that connected pairs of *Tara* Oceans stations. To ensure robustness of our results, we only included pairs of stations that were connected by more than 1000 trajectories. For each pair of stations, $T_{min}$ was defined as the minimum travel time of all trajectories (if any) connecting the two stations.

The source code for ocean circulation simulations can be found at https://mitgcm.org/source-code/, which contains installation instructions for the GitHub repository available at https://github.com/MITgcm/MITgcm; *Losch, 2022*. General configuration information for the high-resolution global calculation can be found in the CVS directory of user contributions: http://wwwcvs.mitgcm.org/viewvc/MITgcm/MITgcm_contrib/ under 'hi_res_cube'. The specific simulation configuration we used followed (*Clayton et al., 2013*) and is available at http://wwwcvs.mitgcm.org/viewvc/MITgcm/MITgcm_contrib/high_res_cube/README.cs510?view=log. The travel time matrix we produced (measured in years) is available as Supplementary Table 15. Standard minimum geographic distance without traversing land (*Rattray et al., 2016*) is available as Supplementary Table 16.

## Correlations of β-diversity, $T_{min}$, and environmental parameters

Our correlation analyses were restricted to *Tara* Oceans samples collected at the surface and did not include DCM samples. We excluded stations that were not from open ocean locations from correlation analyses to avoid sites impacted by coastal processes (those numbered 54, 61, 62, 79, 113, 114, 115, 116, 117, 118, 119, 120, and 121). In analyses where Southern Ocean (including Antarctic) stations were considered independently from other stations, the following were considered Southern Ocean (including Antarctic) stations: 82, 83, 84, 85, 86, 87, 88, and 89. We calculated rank-based Spearman correlations between β-diversity, $T_{min}$, and environmental parameters (either differences in temperature or the Euclidean distance composed of differences in $NO_3 + NO_2$, $PO_4$, and Fe, see above) for surface samples with a Mantel test with 1000 permutations and a nominal significance threshold of $p < 0.01$. For the correlations presented in *Figures 2a and 3* and *Figure 3—figure supplement 1*, correlation values were derived from pairs of stations connected by $T_{min}$ up to the value on the x-axis. We calculated partial correlations of metagenomic and OTU dissimilarity and $T_{min}$ by controlling for differences in temperature and for differences in nutrient concentrations, and partial correlations of dissimilarity with temperature or nutrient variation by controlling for $T_{min}$. We calculated rank-based Spearman partial correlations using the standard formula for two variables x1 and x2 and a controlling variable x3: (cor[x1,x2] − cor[x1,x3] × cor[x1,x3])/(sqrt[1 − cor(x1,x3)^2] × sqrt[1 − cor(x2,x3)^2]).

## Community turnover in the North Atlantic

Tara Oceans stations numbered 72, 76, 142, 143, 144, and all stations from 146 to 151 were located along the main current system connecting South Atlantic and North Atlantic oceans and continuing to the strait of Gibraltar. In addition, we included stations 4, 7, 18, and 30 located on the main current system in the Mediterranean Sea (*Figure 2—figure supplement 2*). As the *Tara* Oceans samples within the subtropical gyre of the North Atlantic and in the Mediterranean Sea were all collected in winter, seasonal variations should not play a role in the variability in community composition that we observed (see Supplementary Table 2). We calculated genomic e-folding times (the time after which the detected genomic similarity between plankton communities changes by 63%) over scales from months to years based on an exponential fit of metagenomic dissimilarity to $T_{min}$ with the form $y = C_0 e^{-x/\tau}$ (where $C_0$ is a constant and $\tau$ is the folding time). Exponential fits for size fractions 0–0.2 μm and 5–20 μm were not calculated due to an insufficient number of sampled stations in the North Atlantic (Appendix 6).

The synthetic map (*Figure 2—figure supplement 2a*) was generated with the R packages maps_3.0.0.2, mapproj 1.2.4, gplots_2.17.0, and mapplots_1.5. We derived dynamic sea surface height from the *Csiro Atlas of Regional Seas* (CARS2009, http://www.cmar.csiro.au/cars) using the hgt2000_cars2009a.nc file and plotted with the R package RNetCDF.

## Imaging methods

Plankton were also collected using WP2 (200 μm mesh) nets, using vertical tows (0–100 m), and preserved with borax-buffered formaldehyde. Taxonomic classification was performed using the ZooScan imaging system (*Gorsky et al., 2010*) and identified with an automatic recognition algorithm to the finest possible taxonomic resolution using Ecotaxa (*Picheral et al., 2017*) . The resulting identifications were manually visualized by taxonomic specialists and either validated or corrected. Resolution of the taxonomic identifications depended on morphological heterogeneity within taxonomic groups. Hence, identifications reached different taxonomic levels, from species to phylum, and most of them reached family level. All images and their taxonomic assignation are accessible within Ecotaxa (https://ecotaxa.obs-vlfr.fr/prj/377). Since all genomic data were collected during day time, we restricted our analysis on day-collected samples. We also discarded non-living objects in our analyses. We estimated β-diversity by calculating Bray-Curtis dissimilarities between pairs of stations based on the relative abundances of each annotated taxonomic unit. Bray-Curtis dissimilarities are available as Supplementary Table 17.

## MAGs analysis

MAG relative abundances in metagenomic samples were retrieved from *Delmont et al., 2022a*. β-diversity was estimated by calculating the Bray-Curtis dissimilarities between pairs of stations based on the relative abundances of each of the 713 MAGs calculated by read mapping in the metagenomes of size fraction 20–180 μm (the size fraction in which MAGs recruit the largest relative share of all reads). We represented PCoA-RGB color of the Bray-Curtis dissimilarity matrix for each sample on a world map (*Figure 1—figure supplement 4g*) following the methodology described above. The Spearman $\rho$ correlation coefficient was calculated between MAG-based β-diversity and metagenomic-based β-diversity from the size fraction 20–180 μm. MAG abundances for the 20–180 μm size fraction are available as Dataset 4. MAG-derived Bray-Curtis dissimilarities for the 20–180 μm size fraction are available as Supplementary Table 18.

## Estimates of percentages of prokaryote reads in eukaryote-enriched size fractions, and vice versa

We used MAG read mappings from *Delmont et al., 2022a*, *Delmont et al., 2022b* to calculate percentages of prokaryote reads in eukaryote-enriched size fractions, and vice versa. For each eukaryote and prokaryote MAG and for each sample, we used the proportion of reads unambiguously mapped to the MAG. For each sample, we next obtained estimates of the percentages of prokaryote or eukaryote reads by summing these relative counts for all prokaryote (bacterial or archaeal) or all eukaryote MAGs. We discarded samples for which both prokaryote and eukaryote relative counts had a zero or negligible sum (<1% of total reads). For the remaining 538 samples, the average percentage represents the average across all samples within a given size fraction.

## Acknowledgements

We acknowledge Oliver Jahn and Mick Follows for providing numerical simulations of particle trajectories from *Tara* Oceans stations. We also acknowledge Stéphane Audic for assistance with metabarcoding analyses, Claude Scarpelli for support in high-performance computing, Mathieu Raffinot and Dominique Lavenier for discussions on sequence comparison algorithms, Samuel Chaffron for help with sample contextual data, Noan Le Bescot (Ternog Design) for assistance in preparing figures, and Marion Gehlen. We thank all members of the *Tara* Oceans consortium for maintaining a creative environment and for their constructive criticism.

We thank the commitment of the following people and sponsors who made this expedition possible: CNRS (in particular Groupement de Recherche GDR3280), European Molecular Biology Laboratory (EMBL), Genoscope/CEA, Fund for Scientific Research – Flanders, VIB, Stazione Zoologica Anton Dohrn, UNIMIB, Paris Sciences et Lettres (PSL) Research University (ANR-11-IDEX-0001–02), the French Government ANR (projects FRANCE GENOMIQUE/ANR-10-INBS-09, MEMO LIFE/ANR-10-LABX-54, POSEIDON/ANR-09-BLAN-0348, PROMETHEUS/ANR-09-PCS-GENM-217, MAPPI/ANR-2010-COSI-004, TARA-GIRUS/ANR-09-PCS-GENM-218), US NSF grant DEB-1031049, FWO, BIO5, Biosphere 2, Agnès b., the Veolia Environment Foundation, Région Bretagne, World Courier, Illumina, Cap L'Orient, the EDF Foundation EDF Diversiterre, FRB, the Prince Albert II de Monaco Foundation, Etienne Bourgois, the *Tara* schooner and its captain and crew. We thank MERCATOR-CORIOLIS and ACRI-ST for providing daily satellite data during the expedition. The bulk of genomic computations were performed using the Airain HPC machine provided through GENCI- [TGCC/CINES/IDRIS] (grants t2011076389, t2012076389, t2013036389, t2014036389, t2015036389 and t2016036389). We are also grateful to the French Ministry of Foreign Affairs for supporting the expedition and to the countries who granted us sampling permissions. *Tara* Oceans would not exist without continuous support from 23 institutes (http://oceans.taraexpeditions.org).

This article is contribution number 136 of *Tara* Oceans.

## Additional information

### Group author details

**Tara Oceans Coordinators**

**Silvia G Acinas**: Department of Marine Biology and Oceanography, Institut de Ciències del Mar (ICM), CSIC, Barcelona, Spain; **Peer Bork**: Structural and Computational Biology, European Molecular Biology Laboratory, Heidelberg, Germany; Yonsei Frontier, Yonsei University, Seoul, Republic of Korea; Department of Bioinformatics, Biocenter, University of Würzburg, Würzburg, Germany; **Emmanuel Boss**: School of Marine Sciences, University of Maine, Orono, United States; **Chris Bowler**: Research Federation for the study of Global Ocean systems ecology and evolution, FR2022/Tara GOSEE, Paris, France; Institut de Biologie de l'Ecole Normale Supérieure (IBENS), Ecole Normale Supérieure, CNRS, INSERM, Université PSL, Paris, France; **Guy Cochrane**: European Molecular Biology Laboratory, European Bioinformatics Institute (EMBL-EBI), Wellcome Trust Genome Campus, Hinxton, Cambridge, United Kingdom; **Colomban de Vargas**: Research Federation for the study of Global Ocean systems ecology and evolution, FR2022/Tara GOSEE, Paris, France; Sorbonne Université, CNRS, Station Biologique de Roscoff, UMR7144, ECOMAP, Roscoff, France; **Gabriel Gorsky**: Research Federation for the study of Global Ocean systems ecology and evolution, FR2022/Tara GOSEE, Paris, France; Sorbonne Universités, CNRS, Laboratoire d'Oceanographie de Villefranche, LOV, Villefranche-sur-Mer, France; **Nigel Grimsley**: CNRS, UMR 7232, BIOM, Avenue Pierre Fabre, Banyuls-sur-Mer, France; Sorbonne Universités Paris 06, OOB UPMC, Avenue Pierre Fabre, Banyuls-sur-Mer, France; **Lionel Guidi**: Research Federation for the study of Global Ocean systems ecology and evolution, FR2022/Tara GOSEE, Paris, France; Sorbonne Universités, CNRS, Laboratoire d'Oceanographie de Villefranche, LOV, Villefranche-sur-Mer, France; **Pascal Hingamp**: Aix Marseille Univ., Université de Toulon, CNRS, IRD, MIO UM 110, 13288, Marseille, France; **Daniele Iudicone**: Stazione Zoologica Anton Dohrn, Villa Comunale, Naples, Italy; **Olivier Jaillon**: Research Federation for the study of Global Ocean systems ecology and evolution, FR2022/Tara GOSEE, Paris, France; Génomique Métabolique, Genoscope, Institut de Biologie François Jacob, CEA, CNRS, Université Evry, Université Paris-Saclay, Evry, France; **Stefanie Kandels**: Structural and Computational Biology, European Molecular Biology Laboratory, Heidelberg, Germany; Directors' Research European Molecular Biology Laboratory Meyerhofstr, Heidelberg, Germany; **Lee Karp-Boss**: School of Marine Sciences, University of Maine, Orono, United States; **Eric Karsenti**: Research Federation for the study of Global Ocean systems ecology and evolution, FR2022/Tara GOSEE, Paris, France; Directors' Research European Molecular Biology Laboratory Meyerhofstr, Heidelberg, Germany; Ecole Normale Supérieure, PSL Research University, Institut de Biologie de l'Ecole Normale Supérieure (IBENS), Paris, France; **Fabrice Not**: Research Federation for the study of Global Ocean systems ecology and evolution, FR2022/Tara GOSEE, Paris, France;

Sorbonne Université, CNRS, Station Biologique de Roscoff, UMR7144, ECOMAP, Roscoff, France; **Hiroyuki Ogata**: Institute for Chemical Research, Kyoto University, Gokasho, Kyoto, Japan; **Stéphane Pesant**: MARUM, Center for Marine Environmental Sciences, University of Bremen, Bremen, Germany; PANGAEA, Data Publisher for Earth and Environmental Science, University of Bremen, Bremen, Germany; **Jeroen Raes**: Department of Microbiology and Immunology, Rega Institute, KU Leuven, Leuven, Belgium; VIB Center for Microbiology, Leuven, Belgium; **Christian Sardet**: Research Federation for the study of Global Ocean systems ecology and evolution, FR2022/Tara GOSEE, Paris, France; Sorbonne Universités, UPMC Université Paris 06, CNRS, Laboratoire d'oceanographie de Villefranche (LOV), Observatoire Océanologique, Villefranche-sur-Mer, France; **Mike Sieracki**: National Science Foundation, Arlington, United States; Bigelow Laboratory for Ocean Sciences East Boothbay, Boothbay, United States; **Sabrina Speich**: Laboratoire de Physique des Océans, UBO-IUEM, Place Copernic, Plouzané, France; Department of Geosciences, Laboratoire de Météorologie Dynamique (LMD), Ecole Normale Supérieure, Paris Cedex, France; **Lars Stemmann**: Research Federation for the study of Global Ocean systems ecology and evolution, FR2022/Tara GOSEE, Paris, France; Sorbonne Universités, CNRS, Laboratoire d'Oceanographie de Villefranche, LOV, Villefranche-sur-Mer, France; **Matthew B Sullivan**: Department of Microbiology, The Ohio State University, Columbus, United States; EMERGE Biology Integration Institute, The Ohio State University, Columbus, United States; Center of Microbiome Science, The Ohio State University, Columbus, United States; Department of Civil, Environmental and Geodetic Engineering, The Ohio State University, Columbus, United States; **Shinichi Sunagawa**: Structural and Computational Biology, European Molecular Biology Laboratory, Heidelberg, Germany; Institute of Microbiology, Department of Biology, ETH Zurich, Vladimir-Prelog-Weg, Zurich, Switzerland; **Patrick Wincker**: Research Federation for the study of Global Ocean systems ecology and evolution, FR2022/Tara GOSEE, Paris, France; Génomique Métabolique, Genoscope, Institut de Biologie François Jacob, CEA, CNRS, Université Evry, Université Paris-Saclay, Evry, France

**Competing interests**

Tara Oceans Coordinators: The other authors declare that no competing interests exist.

**Funding**

| Funder | Grant reference number | Author |
| --- | --- | --- |
| Agence Nationale de la Recherche | HYDROGEN/ANR-14-CE23-0001 | Gaëtan Benoit, Tom O Delmont, Romain Narci |
| Agence Nationale de la Recherche | OCEANOMICS/ANR-11-BTBR-0008 | Nicolas Henry, Cédric Berney, Sarah Romac, Colomban de Vargas |
| National Science Foundation | OCE-1536989 | Jennifer R Brum, Matthew B Sullivan, Simon Roux |
| European Commission | MicroB3/287589 | Antonio Fernàndez-Guerra |
| European Research Council | INMARE/634486 | Antonio Fernàndez-Guerra |
| Commissariat à l'Énergie Atomique et aux Énergies Alternatives | Graduate Student Fellowship | Paul Frémont |
| Graphene Flagship | RITMARE | Daniele Iudicone, Maurizio Ribera d'Alcalà |
| Premiale | MIUR NEMO | Maurizio Ribera d'Alcalà, Daniele Iudicone |
| Conseil Régional de Bretagne | Postdoctoral Fellowship | Daniel J Richter |
| Institute for Bioengineering of Catalonia | Beatriu de Pinós Postdoctoral Fellowship | Daniel J Richter |

| Funder | Grant reference number | Author |
| --- | --- | --- |
| "la Caixa" Foundation | LCF/BQ/PI19/11690008 | Daniel J Richter |
| European Research Council | 949745 | Daniel J Richter |
| National Science Foundation | OCE-1829831 | Matthew B Sullivan |
| Gordon and Betty Moore Foundation | 3709 | Matthew B Sullivan |
| Ohio Super Computer Center | HPC support | Matthew B Sullivan |

The funders had no role in study design, data collection and interpretation, or the decision to submit the work for publication.

## Author contributions

Daniel J Richter, Conceptualization, Data curation, Formal analysis, Methodology, Visualization, Writing – original draft, Writing – review and editing, Investigation, Software; Romain Watteaux, Thomas Vannier, Conceptualization, Formal analysis, Methodology, Visualization, Writing – original draft, Writing – review and editing, Data curation, Investigation, Software; Jade Leconte, Formal analysis, Visualization, Writing – review and editing, Investigation; Paul Frémont, Formal analysis, Visualization, Writing – review and editing, Investigation, Software; Gabriel Reygondeau, Damien Eveillard, Shinichi Sunagawa, Formal analysis, Writing – review and editing; Nicolas Maillet, Nicolas Henry, Formal analysis, Visualization; Gaëtan Benoit, Claire Lemaitre, Methodology, Investigation; Ophélie Da Silva, Formal analysis, Investigation, Writing – review and editing; Tom O Delmont, Investigation, Resources, Writing – review and editing; Antonio Fernàndez-Guerra, Samir Suweis, Formal analysis, Visualization, Writing – review and editing; Romain Narci, Mahendra Mariadassou, Funding acquisition, Writing – review and editing; Cédric Berney, Formal analysis; Frederick Gavory, Eric Mahieu, Investigation; Lionel Guidi, Data curation, Writing – review and editing; Karine Labadie, Investigation, Methodology, Resources; Julie Poulain, Data curation, Investigation, Project administration, Resources; Sarah Romac, Céline Dimier, Sarah Searson, Jennifer R Brum, Resources; Simon Roux, Resources, Writing – review and editing; Stefanie Kandels, Project administration; Marc Picheral, Data curation, Investigation, Resources; Tara Oceans Coordinators, Funding acquisition, Supervision, Writing – review and editing; Stéphane Pesant, Data curation; Jean-Marc Aury, Data curation, Investigation, Software; Eric Pelletier, Chris Bowler, Investigation, Writing – review and editing; Peer Bork, Lee Karp-Boss, Matthew B Sullivan, Writing – review and editing; Fabien Lombard, Data curation, Formal analysis, Investigation, Writing – review and editing, Resources; Eric Karsenti, Supervision, Writing – review and editing; Ian Probert, Writing – original draft, Writing – review and editing; Pierre Peterlongo, Methodology, Investigation, Writing – review and editing; Patrick Wincker, Supervision, Writing – review and editing, Funding acquisition; Colomban de Vargas, Conceptualization, Software, Writing – original draft, Writing – review and editing; Maurizio Ribera d'Alcalà, Conceptualization, Formal analysis, Supervision, Writing – original draft, Writing – review and editing; Daniele Iudicone, Conceptualization, Formal analysis, Methodology, Supervision, Writing – original draft, Writing – review and editing, Funding acquisition, Investigation; Olivier Jaillon, Conceptualization, Formal analysis, Methodology, Visualization, Writing – original draft, Writing – review and editing, Funding acquisition, Investigation, Supervision

## Author ORCIDs

Daniel J Richter http://orcid.org/0000-0002-9238-5571
Nicolas Henry http://orcid.org/0000-0002-7702-1382
Antonio Fernàndez-Guerra http://orcid.org/0000-0002-8679-490X
Cédric Berney http://orcid.org/0000-0001-8689-9907
Damien Eveillard http://orcid.org/0000-0002-8162-7360
Lionel Guidi http://orcid.org/0000-0002-6669-5744
Sarah Searson http://orcid.org/0000-0002-4721-0027
Jean-Marc Aury http://orcid.org/0000-0003-1718-3010
Shinichi Sunagawa http://orcid.org/0000-0003-3065-0314
Chris Bowler http://orcid.org/0000-0003-3835-6187

Ian Probert http://orcid.org/0000-0002-1643-1759
Colomban de Vargas http://orcid.org/0000-0002-6476-6019
Olivier Jaillon http://orcid.org/0000-0002-7237-9596

**Decision letter and Author response**
Decision letter https://doi.org/10.7554/eLife.78129.sa1
Author response https://doi.org/10.7554/eLife.78129.sa2

## Additional files

### Supplementary files
• Transparent reporting form

### Data availability
The authors declare that all data reported herein are fully and freely available from the date of publication, with no restrictions, and that all of the samples, analyses, publications, and ownership of data are free from legal entanglement or restriction of any sort by the various nations in whose waters the Tara Oceans expedition sampled. Metagenomic and metabarcoding sequencing reads have been deposited at the European Nucleotide Archive under accession numbers provided in Supplementary Table 1. Contextual metadata of Tara Oceans stations are available in Supplementary Table 2. Metagenomic dissimilarity, OTU community dissimilarity, imaging community dissimilarity, simulated travel times, geographic distances and MAG dissimilarity are provided in Supplementary Tables 3-18. All Supplementary Tables, in addition to Datasets 1-4 (tables of 18S V9 barcodes and OTUs, the V9 reference database and MAG abundances) are available on FigShare at the following URL: https://doi.org/10.6084/m9.figshare.11303177. Images and their taxonomic assignations are accessible within Ecotaxa (https://ecotaxa.obs-vlfr.fr/prj/377).

The following dataset was generated:

| Author(s) | Year | Dataset title | Dataset URL | Database and Identifier |
| --- | --- | --- | --- | --- |
| Richter D, Watteaux R, Vannier T, Leconte J, Frémont P, Reygondeau G, Maillet N, Henry N, Benoit G, Ophélie DS, delmont t, Fernàndez-Guerra A, Suweis S, Narci R, Berney C, Eveillard D, Gavory F, Guidi L, Labadie K, Mahieu E, Poulain J, Romac S, Roux S, Dimier C, Kandels S, Picheral M, Searson S, Coordinators TO, Pesant S, Aury JM, Brum JR, Lemaitre C, Pelletier E, Bork P, Sunagawa S, Lombard F, Karp-Boss L, Bowler C, Sullivan MB, Karsenti E, Mariadassou M, Probert I, Peterlongo P, Wincker P, Vargas Cd, d'Alcalà MR, Iudicone D, Jaillon O | 2021 | Data from: Genomic evidence for global ocean plankton biogeography shaped by large-scale current systems | https://doi.org/10.6084/m9.figshare.11303177 | figshare, 10.6084/m9.figshare.11303177 |

The following previously published datasets were used:

| Author(s) | Year | Dataset title | Dataset URL | Database and Identifier |
|---|---|---|---|---|
| Tara Oceans Consortium | 2016 | sequence data corresponding to the V9 loop of the 18S rDNA in 850 size-fractionnated plankton communities sampled at 123 location including samples from the mesopelagic zone | https://www.ncbi.nlm.nih.gov/sra/?term=PRJEB16766 | NCBI Sequence Read Archive, PRJEB16766 |
| Tara Oceans Consortium | 2013 | Shotgun Sequencing of Tara Oceans DNA samples corresponding to size fractions for protist | https://www.ncbi.nlm.nih.gov/sra/?term=PRJEB4352 | NCBI Sequence Read Archive, PRJEB4352 |
| Tara Oceans Consortium | 2013 | Shotgun Sequencing of Tara Oceans DNA samples corresponding to size fractions for prokaryotes | https://www.ncbi.nlm.nih.gov/sra/?term=PRJEB1787 | NCBI Sequence Read Archive, PRJEB1787 |
| Tara Oceans Consortium | 2013 | Shotgun Sequencing of Tara Oceans DNA samples corresponding to size fractions for viruses | https://www.ncbi.nlm.nih.gov/sra/?term=PRJEB4419 | NCBI Sequence Read Archive, PRJEB4419 |

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

# Appendix 1

## Comparison of metagenomes and OTUs

Metagenomic comparisons reflect fine-scale differences in genome content at the community level as a function of diversity, genome size, and organismal abundance, and also depend on the rate of evolution of each specific lineage. With exhaustive sampling, metagenomic dissimilarity could theoretically distinguish among genomes in a sample separated by a single mutation. However, our metagenomic sequencing level was likely not able to reach saturation due to the number of genomes per sample and their putative large size (metatranscriptomes, which contain fewer sequences per species than do metagenomes, did not reach saturation within *Tara* Oceans samples *Carradec et al., 2018*). For example, if for a pair of samples we sequence 50% of the total amount of the unique genomic DNA present, we expect the maximum similarity of the two samples to be roughly 25% (0.5 × 0.5). Therefore, the pairwise metagenomic dissimilarities we calculated between samples probably reflected a combination of genomic differences weighted toward more abundant organisms. In contrast, OTUs, obtained by sequencing single marker genes, approach biodiversity saturation (*de Vargas et al., 2015*; *Roux et al., 2016*; *Sunagawa et al., 2015*). However, OTU resolution depends on the choice of the marker to be used, the threshold of similarity for the marker, and its lineage-specific substitution rate, and may therefore confound evolutionarily and/or ecologically distant organisms (*Piganeau et al., 2011*; *Seeleuthner et al., 2018*; *Vannier et al., 2016*; *Worden et al., 2009*; *Wu et al., 2015*). We observed a significant agreement between the two proxies (*Figure 1—figure supplement 2*), although dissimilarities based on OTUs were generally lower than those computed from metagenomic data (*Figure 1—figure supplement 3* a).

Analyses of plankton biogeography produced consistent results based on metagenomic and OTU data (*Figure 1—figure supplement 4*, *Figure 1—figure supplement 5*, *Figure 2—figure supplement 1*, *Figure 3—figure supplement 1*). For simplicity, in the main text, we chose to highlight results based on metagenomes rather than on OTUs for three reasons. First, the metagenomic sequencing protocol and subsequent measurement of dissimilarity were uniform across size fractions, whereas OTUs were defined differently for the viral-enriched, bacterial-enriched and eukaryote-enriched size fractions (Materials and methods). Second, the biogeographical patterns we obtained (see below) may be more evident in comparisons among metagenomic sequences (our data source in identifying genomic provinces), as genomes accumulate single-base changes and other variants more quickly than a single ribosomal gene marker. Third, β-diversity estimated by metagenomic dissimilarity generally displayed higher correlation values with minimum travel time ($T_{min}$; *Figure 2—figure supplement 1*).

## Appendix 2

### Robustness of genomic provinces

We assessed the robustness of genomic provinces in five separate ways. First, we tested five different hierarchical clustering algorithms from the R-package pvclust_1.3–2 (*Suzuki and Shimodaira, 2006*; UPGMA; McQuitty's method; Complete linkage; Ward's method; Single linkage) on the metagenomic pairwise dissimilarities produced by Simka separately for the six organismal size fractions, followed by multiscale bootstrap resampling. We used the cophenetic correlation coefficient from the R-package dendextend_1.5.2 (*Galili, 2015*) to measure how accurately the dendrograms produced by each method preserved the pairwise distances within the input dissimilarity matrices (*Sneath and Sokal, 1973*; *Sokal and Rohlf, 1962*). The ranking of the cophenetic correlation coefficient for different clustering methods within each size fraction (Supplementary Table 19) was consistent with a published large-scale methodological comparison of clustering methods for biogeography, which considered UPGMA agglomerative hierarchical clustering to have consistently the best performance (*Kreft and Jetz, 2010*). Second, we compared clustering results among all size fractions using Baker's Gamma Index (*Baker and Hubert, 1975*) from the R-package corrplot_0.77 (*Wei and Simko, 2016*), which is a measure of association (similarity) between two trees based on hierarchical clustering (dendrograms). The Baker's Gamma Index is defined as the rank correlation between the stages at which pairs of objects combine in each of the two trees. For each type of correlation, the UPGMA was consistently the most correlated with other clustering methods (Supplementary Table 20). This allowed us to conclude, in agreement with previous results (*Kreft and Jetz, 2010*), that the UPGMA method is likely more robust than the other methods we tested.

Third, we compared the genomic provinces found by our UPGMA hierarchical clustering approach to those found by two different non-hierarchical methods: K-means on the positions found by multidimensional scaling and spectral clustering on the nearest-neighbor graph. Both methods rely on (i) a dissimilarity matrix and (ii) a tuning parameter (dimension of the projection space for K-means, and number of neighbors for spectral clustering). K-means uses the numeric values of the dissimilarities, whereas spectral relies only on their ordering (e.g. community A is closer to B than to C). We compared the genomic provinces to clusters found by K-means and spectral clustering for all values of the tuning parameter using the Rand Index (RI; from the GARI function of the loe R package version 1.1 *Terada and Luxburg, 2016*), a score of agreement between partitions. Results are reported as mean ± SD of the RI: 1 means perfect agreement and 0 complete disagreement. Fourth, in order to assess the significance of the genomic provinces, we performed a multivariate ANOVA to partition metagenomic dissimilarity across regions, using the adonis function of the vegan R package version 2.5–4 (*Oksanen et al., 2019*). Note, however, that since the same data were used both to construct the genomic provinces and to assess their significance, the p-values estimated by ADONIS might be anti-conservative. The results of the third and fourth analyses are presented in Supplementary Table 21.

Fifth, we found that clustering of samples in genomic provinces was consistent with a complementary visualization based on the same data: RGB colors derived from the first three axes of a principal coordinates analysis (PCoA-RGB) of β-diversity, in which similar colors represent similar communities (*Figure 1—figure supplement 4*; see Methods). Samples within the same genomic province generally shared the same range of PCoA-RGB colors. Because the clustering approach was hierarchical, samples sharing some similarity could have been assigned to different genomic provinces due to binary decisions during the clustering process. This was also reflected in the PCoA-RGB colors, where the boundaries of genomic provinces did not indicate a complete change of communities among genomic provinces (and, conversely, belonging to the same genomic province did not imply identical communities). Nonetheless, samples with similar PCoA-RGB colors were generally situated in closely-related branches in the UPGMA tree (*Figure 1—figure supplement 6*). An illustrative example is genomic province F5 (of the 180–2000 μm size fraction; *Figure 1—figure supplement 4f*), which encompassed stations in the Atlantic, Mediterranean Sea and some subtropical stations in the Indo-Pacific. In this wide region, the PCoA-RGB colors indicate the variation in community composition within the genomic province, and also reflect the relatedness of F5 to its adjacent samples, in particular those in the subtropical Atlantic/Pacific region F4, its neighbor in the UPGMA tree (*Figure 1—figure supplement 6f*).

## Appendix 3

## Comparison of genomic provinces to previous biogeographical divisions

Current approaches in biogeographic theory divide the ocean into regions based either on expert knowledge applied to satellite data, as in the hierarchical nesting by *Longhurst, 2006* into biomes (macroscale, essentially representing a division of the world's oceans into cold and warm waters, and coastal upwelling zones) and biogeochemical provinces (BGCPs, areas within biomes defined by observable boundaries and predicted ecological characteristics), or, alternatively, into the objective provinces of *Oliver and Irwin, 2008*, which are based solely on statistical analyses. Longhurst BGCPs are based upon, primarily, monthly variations of chlorophyll *a*, the geography of the seasonal cycle of physical factors (such as the depth of the upper ocean mixed layer) and surface temperatures. In turn, these ocean properties are strongly modulated by oceanic currents (e.g. moderate to large mixed layer depths are observed generally on the poleward side of the subtropical gyres). In contrast, the objective global ocean biogeographic provinces proposed by *Oliver and Irwin, 2008* were based upon clustering temporal variability of chlorophyll concentration and surface temperatures, both measured from satellite data. They combined a proxy for the intensity of primary productivity with water temperature, therefore emphasizing regions similar in their temporal variability for both properties (which essentially corresponds to the seasonal cycle). None of these ocean partitions directly considered organismal community composition.

We tested whether genomic provinces were comparable with these partitions by performing an analysis of similarity (ANOSIM; *Figure 1—figure supplement 4a-f*, insets; Methods). The four small size classes, 0–0.2 μm, 0.22–1.6/3 μm, 0.8–5 μm, and 5–20 μm (*Figure 1—figure supplement 4a-d*) were more consistent with Longhurst BGCPs. In contrast, for the two larger size fractions 20–180 μm and 180–2000 μm, the three biogeographical divisions were not strongly different within the ANOSIM (*Figure 1—figure supplement 4e-f*).

From an oceanographic point of view, plankton should be quasi-neutrally redistributed (i.e. homogenized) by currents and their biogeography should follow the structure of the main recirculations, within a range of physiologically compatible temperatures. In this point of view, our results are consistent with the large-scale geographic distributions found by *Hellweger et al., 2014* using a neutral model.

Although our analyses of the influence of transport and other environmental conditions on plankton biogeography are focused on surface samples (the depth at which our simulations of transport was conducted), we included samples collected at the DCM in our genomic provinces in order to produce a broader description of plankton biogeography. In general, samples at the surface and DCM from the same station clustered together in the same genomic provinces, although we observed a minority of cases in which surface and DCM samples from the same station were found in different genomic provinces, and even a few genomic provinces composed only of DCM samples (e.g. C6 in the 0.8–5 μm size fraction; *Figure 1—figure supplement 4*).

## Appendix 4

### Differences in genomic province sizes among organismal size fractions

Globally, we obtained more numerous, smaller genomic provinces in the smaller size fractions and fewer, larger genomic provinces in the larger size fractions (*Figure 1—figure supplement 4*, *Figure 1—figure supplement 7*). We observed a similar pattern using OTU data (*Figure 1—figure supplement 5*). Whereas smaller size fractions generally lacked geographically widespread genomic provinces containing numerous *Tara* Oceans samples, the two largest size fractions were both characterized by two very widespread genomic provinces: F5 and F8 for the 180–2000 μm size fraction, and E5 and E6 for the 20–180 μm size fraction. These large genomic provinces were latitudinally limited by the boundary between the subtropics and subpolar regions, and spanned different oceanic basins. Notably, in the southern hemisphere the subtropical gyres actually form a single supergyre (*Speich et al., 2007*) and there are almost no metabolic (mainly temperature) barriers between the northern and southern subtropical gyres (see *Figure 1—figure supplement 4*), potentially explaining genomic provinces in the 20–180 μm and 180–2000 μm size fraction that contain samples from the North and South Atlantic. For example, in the 180–2000 μm size fraction, F5 mostly covered the North and South Atlantic Oceans and adjacent systems, and F8 covered the Indo-Pacific low- and mid-latitudes. No clear correspondence existed with biogeochemical patterns (e.g. nutrient ratios), except for the clusters coinciding with upwelling systems (F3 for the California upwelling, F7 for the Chile-Peru upwelling, and F2 for the Benguela upwelling system) and for the samples collected at the DCM in the Pacific subtropical gyres (F5); this is consistent with the comparison of genomic provinces to previous biographical divisions, in which the genomic provinces of smaller size fractions were more consistent with Longhurst BGCPs, but those of larger size fractions were not (Appendix 3). A bimodal zooplankton species distribution (split into subtropical and subpolar communities, with ubiquitous warm water species) was also detected by a recent study on copepod population dynamics that used alternative approaches to analyze the same metagenomic data set (*Madoui et al., 2017*) (see their *Figure 2*). More locally, within the North Atlantic (see also Appendix 6), along the northern boundary of the subtropical gyre, cold and warm copepod species overlapped because of cross-current dispersal. Nonetheless, although both cold and warm species appeared to be able to travel long distances, mixing among them was not sufficient to create a local genomic province in our data.

We interpret the difference in genomic province sizes between smaller and larger size fractions as the result of various factors. Plankton smaller than 20 μm (femto-, pico-, and nanoplankton), which represent most of the prokaryotic and eukaryotic phototrophs (*de Vargas et al., 2015*; *Sunagawa et al., 2015*), are sensitive to a suite of environmental factors (i.e. temperature [*Eppley, 1972*], nutrients, and trace elements [*Moore et al., 2013*]; see also *Figure 1—figure supplement 7*) and generally have a shorter life cycle, together leading to faster fluctuations in their relative abundance in the communities we sampled. In contrast, larger plankton have longer life cycles and, if they are predators that are not strongly selective in their feeding, or are photosymbiotic hosts capable of partnering with multiple different symbionts, may cope with local fluctuations in environmental conditions. Therefore, they should be affected primarily by large scale, mostly latitudinal, variations in the environment, leading to larger genomic provinces, whereas smaller plankton are grouped into smaller provinces more influenced by local environmental conditions. Overall, this difference in biogeography suggests a size-based decoupling between smaller and larger plankton (which may also extend to nekton such as tuna and billfish; *Reygondeau et al., 2012*), with implications for the structure and function of oceanic food webs and other types of biotic interactions.

## Appendix 5

### Genomic provinces as stable ecological continua

As plankton communities are transported by ocean currents, they change over time due to the various processes that occur in the context of the seascape: variations in temperature, light and nutrients (where changes in the latter may also be induced by plankton communities), intra- and interindividual and species biological interactions, and mixing with neighboring water masses. Thus, a continuum of composition among nearby samples is expected as a natural consequence of community turnover within the seascape over time. We observed the effects of continuous turnover in our biogeographical analyses (*Figure 1* a, *Figure 1—figure supplement 4*, *Figure 1—figure supplement 5*, Appendix 2) in which nearby samples often reflected gradual, but not complete changes in community composition.

We measured the time window of transport by currents separating two samples during which the changes in their community composition were maximally correlated with travel time, resulting in a global average of $T_{min}$ less than roughly 1.5 years. This represents the travel time during which predictable continuous turnover occurs in our data set. This time scale of 1.5 years is probably an underestimate, since our sparse sampling did not cover all current systems. Notably, $T_{min}$ does not necessarily define the turnover rate itself, which depends on how strongly different seascape processes affect communities with differing biological characteristics (see Appendix 6).

## Appendix 6

### Community turnover in the North Atlantic

In order to characterize the impact of physical and biological processes on changes in metagenomic composition during travel along currents, we focused on the well-known current systems crossing the North Atlantic into the Mediterranean Sea (the Gulf Stream and other currents around the subtropical gyre *Dornelas et al., 2014*; *Fofonoff, 1981*; *Franklin, 1786*; *Talley et al., 2011*; *Figure 2—figure supplement 2a*). Across this region, the piconanoplankton (0.8–5 μm) were split into three genomic provinces, C5, C8, and C3, each less than 5000 km wide (~11 months of travel time; *Figure 1—figure supplement 4c*). In contrast, mesoplankton (180–2000 μm) biogeography corresponded to a single province, F5, spanning from the Caribbean to Cyprus (>9700 km or ~18 months of travel time; *Figure 1—figure supplement 4f*; see also Appendix 4). Metagenomic dissimilarity and $T_{min}$ were strongly correlated within the region (Spearman's $\rho$ between 0.44 and 0.86 depending on size fraction, *Figure 2—figure supplement 2b-e*), which allowed us to explore the relationship of genomic province size, ocean transport, and plankton community turnover over scales from months to years. We calculated metagenomic turnover times as e-folding times based on an exponential fit of metagenomic dissimilarity to $T_{min}$ (ranging from a few months to a few years, Methods). The metagenomic turnover time of smaller plankton (<20 μm) was approximately 1 year. In contrast, for the larger size fractions, the metagenomic turnover time was approximately 2 years, suggesting that a lower turnover rate for larger plankton may explain their geographically larger genomic provinces.

We note that our results on metagenomic turnover time appear different from a recently published study that also calculated turnover rates for plankton, which found faster rates for larger organisms (*Villarino et al., 2018*). This may be explained by two significant differences between our approach and theirs: first, their measurements of β-diversity were based on presence/absence (Jaccard) comparisons among either morphological species or OTUs, whereas our calculations of turnover time above were based on metagenomic sequences. As described above (Appendix 1), there are significant differences in resolution between OTU-based and metagenomic data, and we would expect similar differences in resolution between organismal observation data and metagenomic sequences. In fact, due to these differences in resolution, our estimates of metagenomic turnover based on OTU rather than metagenomic data show a similar trend to those of (*Villarino et al., 2018*; *Figure 2—figure supplement 2f-i*). Second, their turnover rates were calculated separately for individual plankton groups (the nine main groups were prokaryotes, coccolithophores, dinoflagellates, diatoms, all microbial eukaryotes, gelatinous zooplankton, mesozooplankton, macrozooplankton, and myctophids), whereas our metagenomic data represent samples of the full plankton community within each size fraction. Among these, several groups (e.g. dinoflagellates or mesozooplankton) would be expected to be found across multiple *Tara* Oceans size fractions, blurring potential comparisons. Thus, our study and Villarino et al. calculated rates of change using broadly similar approaches, but based on very different underlying biological substrates.

## Appendix 7

## Plankton biogeography is robust to missing samples

Although many individual *Tara* Oceans stations are missing metagenomic or metabarcode sequencing data for a subset of size fractions (*Figure 1—figure supplement 1b*), all oceanic regions have broad coverage for each size fraction, with the exception of the viral-enriched size fraction in the North Atlantic. In fact, the largest source of missing data in our study is due to limited sampling of the viral-enriched size fraction in this region. Nevertheless, we found a pattern for organismal biogeography and for its relation with transport time that is not dependent on the size fraction, and therefore also does not depend on the particular size fractions sampled at specific set of sampling sites. In our analyses, we found a consistently similar patterns across the four smaller size fractions (each fraction was sampled and analyzed independently from the others) as opposed to the two larger ones. In addition, for our results relating to ocean transport time, the fact that the sampling sites are not exactly the same among size fractions actually lends robustness to our results, since it means that the dynamics we found are not overly dependent on any one selected site, region, or subset of sampling stations.

