## [Editor Report]

Richter and colleagues present an impressive analysis of metagenomic, OTU and imaging data collected from >100 ocean locations worldwide, with the purpose of elucidating the role of large-scale currents on global-scale marine plankton biogeography. The topic is exciting and timely.

---

## [Decision Letter]

**Decision letter after peer review:**

[Editors’ note: the authors submitted for reconsideration following the decision after peer review. What follows is the decision letter after the first round of review.]

Thank you for submitting your work entitled "Genomic evidence for global ocean plankton biogeography shaped by large-scale current systems" for consideration by *eLife*. Your article has been reviewed by 3 peer reviewers, and the evaluation has been overseen by a guest Reviewing Editor with expertise in this field, and a Senior Editor. The following individual involved in review of your submission has agreed to reveal their identity: Francisco Rodríguez-Valera (Reviewer #3).

Comments to the Authors:

We are sorry to say that, after consultation with the reviewers, we have decided that your work cannot be considered further in its current form for publication by *eLife*. As you will see in the reviews, the great potential of the paper was generally appreciated, but all reviewers expressed substantial concerns about the methodology and the conclusions that can be drawn from it. It is *eLife*'s policy not to request revisions which we anticipate are likely to take more than two months to complete and thus we are rejecting your submission; however, if you believe that our concerns, outlined below, can be adequately addressed, we encourage you to submit a substantially revised version of your manuscript. If you do decide to re-submit, please include a point-by-point explanation of how these concerns have been addressed.

Richter and colleagues present a large-scale analysis of metagenomic, OTU and imaging data collected from >100 ocean locations worldwide, with the purpose of elucidating the role of large-scale currents on global-scale marine plankton biogeography. The topic is exciting and timely, and should be of broad interest to the fields of marine microbiology, biological oceanography, plankton ecology and physical oceanography. One strong aspect of this study is that it compares community dissimilarities with travel times computed from a global ocean circulation model, rather than simply considering geodesic distances between locations. The data included with this manuscript will also likely be of great value to other research groups. There are, however, substantial concerns regarding the methodology and the conclusions that can be drawn from the work. A specific concern regards the use of 31 bp k-mers for calculating metagenomic dissimilarities and the difficulty of interpreting such dissimilarities biologically, given the many nuances of genome features from viruses to zooplankton. Concerns were also raised regarding the calculation of Tmin based on the surface layer, given that current velocity and direction near the surface and at lower depths of the mixed layer may diverge. Finally, there were questions about the relatively narrow distribution of Tara Oceans samples from similar latitudes and (largely) from the surface, and the fact that different locations were sampled at different seasons (which influence the extent of stratification).

*Reviewer #1:*

1. One strong aspect of this study is that it compares community dissimilarities with travel times computed from a global ocean circulation model, which arguably makes more sense than simply considering geodesic distances between locations (as most previous studies have done). The data included with this manuscript will also be of great value to other research groups.

2. The authors have shown that travel time correlates strongly with metagenomic community dissimilarities (lines 208-209), but they also found strong correlations between travel time and environmental differences (lines 232-233), and correlations between metagenomic differences and environmental differences (Figure 3 and lines 191-192). They also found that their genomic provinces often matched environmental differences. This begs the question to what extent the genomic provinces are caused by circulation within the provinces (which probably "homogenizes" communities within a province) rather than simply environmental differences between provinces (or environmental homogeneity within provinces). In other words, even if there was no circulation (thus communities are entirely determined by local environmental conditions), would we still expect to see the genomic provinces that the authors found? I would have expected a strong discussion on disentangling environmental selection effects from dispersal effects. The authors do marginally touch upon this issue by calculating partial correlations (Supplemental Figure S9), but they don't discuss this at all in the main text. Their Supplemental Figure S9c actually suggests that correlations between metagenomic dissimilarities and environmental differences (temperature and/or nutrients), when controlling for Tmin, are often stronger than correlations between metagenomic dissimilarities and Tmin (when controlling for temperature and nutrients). This might suggest that for some size classes genomic provinces are mostly caused by environmental homogeneity within basins/provinces, rather than the homogenizing action of currents. As it stands, I find the take-home messages of the paper rather vague and inconclusive from a mechanistic (i.e., rather than descriptive) point of view.

3. The authors calculate metagenomic dissimilarities with Simka based on counts of 31bp-k-mers (instead of, say, species counts or gene ortholog counts). While this is computationally efficient, it makes a biological interpretation of their metagenomic β-diversity hard. The original Simka paper demonstrated that k-mer distances correlate with taxonomic distance matrixes, but they don't seem to examine the relationship between k-mer dissimilarities and function-centric dissimilarities, i.e., dissimilarities in metabolic functions (as inferred by functionally annotating metagenomic reads or contigs). I suspect that k-mer-based distances may not correlate well with function-based dissimilarities.

Recommendations for the authors

I strongly recommend that the authors provide a quantitative and thorough discussion of my point 2 in their main text.

Related to point 3: Why not also perform an analysis in terms of functional (gene-centric) metagenomic dissimilarities? It seems that the authors only considered 100 million reads per sample anyway, so it should be feasible to annotate those and compare KO tables between samples. This would aid in the ecological interpretation of their findings. It would also facilitate comparison to other recent function-focused studies of marine microbial biogeography, such as [Ramírez-Flandes et al. 2019, DOI:10.1073/pnas.1817554116] or [Coles et al. 2017, DOI:10.1126/science.aan5712].

*Reviewer #2:*

The authors of the Tara Ocean consortium used this unique metagenomics data set of plankton size classes to test whether global biogeographic patterns exist and how these patterns are affected or even structured by global ocean currents. This data set of six size classes including viruses (<0.2 µm), pico- and nanoplankton (prokaryotes, protists), micro- and macroplankton (protists, metazoans) up to a size of 2 mm is really unique for such an effort. It provides a unified basis by using genomic DNA in all size fractions and thus avoids biases related to methodological differences of how the plankton communities of the different size classes were analyzed. By using station- and satellite-based metadata and correlating the metagenomic data of the stations and their dissimilarities to water mass transport derived from the MITgcm global ocean current model, the authors find global biogeographic patterns. Interestingly, these patterns of the different size classes are positively correlated to the minimum travel time of water masses up to 1.5 years, suggesting that biogeographic provinces evolve and presumably persist over this time span. Beyond, other factors become more important, such as temperature or the nutrient regime. The data also show that the scale of the biogeographic patterns is inversely related to the size classes. Further, a hierarchical cluster analysis of the metagenomics data of the different size classes yielded global genomic provinces which grouped together stations of related environmental and biogeochemical properties beyond water masses.

The statistical evaluation is very rigorous and for various correlations multiple and different analyses have been performed. The pairwise dissimilarity analysis of stations was done both for the metagenomics and OTU data yielding basically similar results and thus confirming the validity of the outcome.

Because of the novelty of this analytical approach this study may become seminal for further similar analyses. In order to become a really solid basis for such future analyses I suggest that the authors should consider the following critical points to further improve the study and to revise the manuscript accordingly.

1. I tried to find specifications for sampling of the larger plankton size classes in this manuscript and other Tara Ocean publications. According to the available information metazoans were also collected from Niskin bottles. If this is correct it means that only rather few organisms per sample and depth were collected, assuming a total abundance of <20 animals per liter in oligotrophic regions. So the questions arises how reliable the data on the larger size classes are for individual taxa compared to the smaller size classes where this point is not an issue. Usually zooplankton is collected by net hauls to obtain enough material. However, if net hauls are used they integrate over sections of the water column and it is impossible to sample a particular depth such as DCM. This point may also be an issue for the cluster analysis of the genomic provinces.

2. The cluster analysis of the genomic provinces shows that at quite a few stations the sample near the surface and at the DCM affiliates to different provinces. For the analysis of the water transport the surface layer of the MITgcm model was used. However it is known that current velocity and direction near the surface and at lower depths of the mixed layer may diverge, in particular towards the equatorial currents (Cravatte et al. 2017, J. Phys. Oceanogr. 47: 2305, DOI: 10.1175/JPO-D-17-0043.1; Hu et al. 2020, Sci. Adv. : eaax7727, DOI: 10.1126/sciadv.aax7727). How would the water transport and plankton dispersal change if only the near surface samples were used for this analysis (which would be the correct way for this analysis) or if this analysis were done with the MITgcm model for the depth section of the DCM (which would imply a reduced number of stations and an adjustment of depths because the depth of the DCM was variable)?

3. Stations in oceanic gyres dominate the sampling grid of the Tara Ocean expedition and stations in coastal and equatorial upwelling regions are greatly underrepresented. Therefore, and based on some discussion in the manuscript, the impression emerges that the biogeographic patterns and their relationship to Tmin is mainly true for oceanic gyres. I suggest that the authors should elaborate on this point and may also consider these constraints in their biogeographic analysis.

4. An important outcome of the study are the different scales of the dispersal of the size classes with Tmin. In a previous publication of the Tara Ocean consortium (Sunagawa et al. 2015, Figure 4B) they show a plot of an increasing dissimilarity of the prokaryotic communities with distance up to appr. 5000 km. In this manuscript the authors mention that they calculated correlations of the dissimilarities with distance for the different size classes but do not show any data. The study would greatly benefit when they show these plots for the different size classes which should yield different patterns. The distance of 5000 km may relate to the travel time of 1.5 years over an oceanic gyre.

In addition further recommendations are as follows:

l. 180-183, 195-199 and other places: There are quite a few genomic provinces of the prokaryotic and protest enriched size classes which go far beyond one ocean basin and even occur at one station near the surface and at the DCM. So be more precise in describing these features. Howe would you cope with genomic provinces which encompass similar stations but in corresponding gyres of the northern and southern hemispheres?

l. 181: delete "to" in this part of the sentence:.….tended to be limited to a single ocean basin and [TO] approximately correspond to…

l. 366: must be "….the same number of reads WAS used…."

l. 1019-1024: You hypothesize that the travel time of 1.5 years is equivalent to the time needed for crossing an oceanic gyre. I assume that this must not remain a hypothesis because I am convinced that ground truth data exist which provide such travel times, may be from the Argo floats program.

*Reviewer #3:*

This is another contribution from the Tara consortium and collaborators, in this case, they try to correlate ocean circulation with plankton biogeography. They have done a number of statistical comparisons using metagenomic data to analyze the effect of a parameter that they call Tmin, the minimal travel time, an estimation of the time that would transfer a water volume from one station to another as deduced from the expected water-mass movement. The problem of this approach (as with previous Tara papers in the view of this reviewer) is the random distribution of Tara stations at similar latitudes and (largely) from the surface and at different seasons. They contemplate the ocean as a two-dimensional system, largely ignoring the third dimension (depth) and the water column stratification that appears at most of these stations seasonally. Surface water movements have been considered without regard to the potentially more important vertical ones that happen when the water column mixes in colder seasons. Thus, their conclusions are flawed by a poor sampling strategy. The authors could have used more structured sampling efforts such as Geotraces, at least to check their overarching conclusions.

A second major flaw of this work is that the main source of information is what they call "metagenomic dissimilarity". It is actually the reciprocal of the ratio of shared (100% identity) 31 nucleotide K-mers between stations out of a pool of several million Illumina reads. This is a quite rough estimate of similarity that does not contemplate the nuances of genome features from virus to zooplankton. For example, the presence of IS or related elements in prokaryotic genomes might bias this parameter strongly, as would the presence of multiple repeats of rRNA genes in eukaryotic cells. The biological significance of metagenomic dissimilarity should be carefully assessed. I do not imply that it cannot be used, but to reach conclusions of such weight ("oceanic genomic provinces") a much more refined sampling strategy and analysis of the data would have been required. For example, why were the myriad of MAGs derived from prokaryotes and viruses at different geographical sites not considered? At least as a control for their claims. Actually, the several reports of nearly identical genomes at different oceanic provinces points towards the opposite. I do not believe the evidence presented here warrants the kind of conclusive statements presented.

In what follows I have identified specific points that would need clarification or modification in case the work had to be published.

Ln 98. There are now many studies on the biogeography of microbes based on metagenomics, including depth profiles and similarities along different transects so this sentence is just wrong.

Ln 102 seascape= metadata

Ln 104 the approach is not consistent (e.g., amplicon sequencing and metagenome similarity)

Ln112 and mixing with deeper layers

Ln118 neighboring (and deeper) water masses

Ln 155 However, some taxa information would have enriched enormously the manuscript

Ln 160 MAG abundance is not a reliable estimate of microbe abundance, often it is the opposite i.e. assembled microbes are not particularly abundant in the environment as exemplified by SAR11 or picocyanobacterial (several references).

Ln 163 explain "significant"

Ln 165 to the end of the paragraph. Extremely subjective i.e., heterogeneous compared to what? A depth profile will show that microbes at a 50 m distance in depth are likely more dissimilar than those located a 500 Km but at the same depth.

Ln 175. Colors are a very subjective representation, different shades of grey or lines of different thickness connecting stations as actually presented in Figure 2 b are easier to interpret.

Ln 187 surface temperature?

Ln 187 temperature or the cognate community?

Ln 190 which size classes?

Ln 194 or those microbial communities vary more sharply with depth and when they upwell (with nutrients) disrupt the water-mass continuity more

Ln 197 and vertical transport?

Ln 219 In any case, it would be interesting to know how dissimilarity correlates with geographic distance as well since Tmin will vary accordingly at shorter distances. It is to be expected that a transect following the Gulf Stream (small Tmin) will show high similarity. This would be a good control of the method of metagenome comparison.

Ln 223 Even more important would be the season and whether the water column is stratified or mixed as would be the case in winter in temperate latitudes (most of Tara samples).

Ln 239 temperature is more correlated with depth and season, particularly at the temperate latitudes.

Figure 2. Many points appear very divergent, can you explain the most extreme cases and label them in the figure?

Ln 332 what happens when the water column is mixed and there is no DCM?

Ln 252 easy enough to pinpoint upwelling areas

Ln 253 There seems to be something wrong with the plots presented in Supplementary Figure 2. If anything, they seem to prove that there is no clear correlation between OTUs and metagenomic dissimilarity what is actually not surprising considering the difficulty when trying to correlate data obtained in so different approaches and with different types of genomes (prokaryotic versus eukaryotic or even multicellular planktonic organisms).

[Editors’ note: further revisions were suggested prior to acceptance, as described below.]

Thank you for resubmitting your work entitled "Genomic evidence for global ocean plankton biogeography shaped by large-scale current systems" for further consideration by *eLife*. Your revised article has been evaluated by Meredith Schuman (Senior Editor), as well as the previous Reviewing Editor and both previous reviewers.

The reviewers and reviewing editor appreciate the extensive work that the authors put into their revised manuscript and the detailed explanations provided in their response letter. There are still some concerns expressed by some of the reviewers, particularly with regards to (a) the ambiguity of using daily sunshine duration as a proxy for seasonality, (b) the potential inclusion of smaller organisms in larger nominal size fractions (for example host-associated microbes may be included in the 180-2000 fraction, thus distorting your analyses), and (c) the focus on surface currents and the omission of the ocean's 3-dimensional structure and variable stratification. The reviewers and reviewing editor have the following minimum recommendations for addressing these issues:

1. Please indicate in one of your maps what sampling sites had a mixed water column (e.g. less than 5ºC difference from surface to the subsurface DCM sample) and which ones had a stratified water column (e.g. more than 5ºC difference between surface and subsurface).

2. Please acknowledge in your paper (e.g., the introduction and discussion) the potential significance of depth, in particular highlighting the point that in the mesopelagic the relationship between composition of plankton communities and currents may be quite different than at the surface.

3. Please acknowledge (e.g. in the introduction or discussion) that the ocean is a tridimensional system in which the main axis of variation is depth, and that a focus on surface currents is a limitation of this study.

4. Please acknowledge in your paper the caveat that daily sunshine duration does not unambiguously map to seasonal effects (since in Spring and Fall daily sunshine durations coincide), and that ocean biology, chemistry and stratification often differ between Fall and Spring.

5. Please acknowledge in your paper that your size fractions are operational, i.e., not necessarily mapping precisely to organism sizes but instead a priori only mapping to "whatever is captured between two specific filter pore sizes". Please also provide some supporting information regarding the fraction of microbial (and perhaps even viral reads) present in the larger nominal size fractions, so that the readers can judge to what extent this may have been an issue.

6. Please clarify in line 143 why only 18S sequences are mentioned and not 16S and correct if necessary.

7. Please also check in line 142 if 24.2 TB of data were indeed analyzed, or if this is the total number of sequences but not all were actually analyzed.

The review from Reviewer #3 provides some additional details regarding the above revisions, which could help you to formulate your response to the essential revisions. The full reviews from all reviewers are provided for your reference below.

*Reviewer #1:*

The authors have clearly put a lot of effort into explaining their reasoning and improving the language of the manuscript, although in most cases they did not actually adjust/extend their analyses to address reviewer concerns. Hence, the paper's conclusions still remain in my view largely descriptive rather than mechanistic, for example, the question on the relative effects of the environment vs dispersal on marine microbial biogeography remains largely unaddressed.

That said, overall I think this is an important paper with a strong dataset, and the community should just take it for what it is.

*Reviewer #2:*

The authors addressed all my concerns and questions satisfactorily and revised the respective parts of the manuscript accordingly.

My impression is that the manuscript gained substantially regarding clarity and limitations of the findings based on all three reviews.

Based on my view of the revised manuscript I recommend its acceptance.

*Reviewer #3:*

There is no way to separate the "6 organismal size fractions" by filtrations. Take for example the 0.22 to 1.6 fraction, although it will be enriched in bacterial DNA up to 20% will be viral. Or even worse, 5 to 20 will have large amounts of virus and bacteria, even 180-2000 "animal" fraction will have all the microbiomes of planktonic animals that will lead to unpredictable background noise in "metagenomic dissimilarity"

Daily sunshine duration cannot be a proxy for season. In fact, autumn and spring days can have a similar duration but are the opposite in terms of stratification, nutrient availability and community structure (think for example of comparing March 21st with September 21st both close to the equinox but extremely divergent in conditions in temperate latitudes).

The use of eukaryotic (even animal) MAGs is too novel (only a preprint yet) and requires extensive benchmarking before it is used to test such a scheme of dividing the world oceans into "genomic provinces". Something similar would have been much more reliable if applied to prokaryotic MAGs.

Finally, Tara Oceans although very large in terms of Terabases is not a good geographical sampling since it did not have into account depth profiles or season of sampling.

---

## [Author Response]

[Editors’ note: the authors resubmitted a revised version of the paper for consideration. What follows is the authors’ response to the first round of review.]

Richter and colleagues present a large-scale analysis of metagenomic, OTU and imaging data collected from >100 ocean locations worldwide, with the purpose of elucidating the role of large-scale currents on global-scale marine plankton biogeography. The topic is exciting and timely, and should be of broad interest to the fields of marine microbiology, biological oceanography, plankton ecology and physical oceanography. One strong aspect of this study is that it compares community dissimilarities with travel times computed from a global ocean circulation model, rather than simply considering geodesic distances between locations. The data included with this manuscript will also likely be of great value to other research groups. There are, however, substantial concerns regarding the methodology and the conclusions that can be drawn from the work. A specific concern regards the use of 31 bp k-mers for calculating metagenomic dissimilarities and the difficulty of interpreting such dissimilarities biologically, given the many nuances of genome features from viruses to zooplankton. Concerns were also raised regarding the calculation of Tmin based on the surface layer, given that current velocity and direction near the surface and at lower depths of the mixed layer may diverge. Finally, there were questions about the relatively narrow distribution of Tara Oceans samples from similar latitudes and (largely) from the surface, and the fact that different locations were sampled at different seasons (which influence the extent of stratification).

We appreciate the interest of the editor and the reviewers in our work, and we thank them for their time in evaluating our manuscript. In our response and in the associated modifications to our manuscript, we believe that we adequately address all of the concerns that they raised. We address each point in detail below as part of our response; here, we present general responses to the three main concerns raised in this summary.

1. The use of 31 bp k-mers

Although the principal focus of our manuscript is the analysis of metagenomes using k-mers, we confirmed all main results with OTUs and with imaging data (both of which provide different levels of taxonomic resolution from one another and from metagenomes) and, for the 20-180 µm size fraction, metagenome-assembled genomes (MAGs).

We are aware that the probability of finding common 31 bp k-mers in a pair of communities depends not only on community composition, but also on other factors such as the size and complexity of their genomes. For example, the presence of abundant species with small genomes would increase the probability of finding a match (see also Figure 1—figure supplement 3b). For this reason, we performed the same analyses using OTUs for all size fractions, resulting in the same principal conclusions. In the main text, we focused on results based on k-mers because we obtained better correlation values for most plankton size fractions, which we hypothesize is due to the higher resolution of genomic data.

In addition to OTUs, we validated our results using imagery data collected for organisms 200 µm and larger. We show that for the larger size fractions, community dissimilarities based on k-mers, imagery and OTUs are well correlated (Figure 1—figure supplement 2). However, the correlation of imagery data with environmental parameters and T_min_ is lower (Figure 3—figure supplement 1). We expected these lower correlations due to the much higher resolution of genomic data (see also Supplementary Information 1 for a similar argument for OTU data). In brief, for the largest size fractions, computing community similarities using imagery, OTUs and k-mers provides different tradeoffs between sampling depth and resolution. Our analysis shows that the three estimates are well correlated, but that k-mers from metagenomes provide better correlations with T_min_ and with variations in environmental conditions, most likely due to their higher resolution.

We also believe that this empirical demonstration of using k-mers to highlight Lagrangian variation of plankton communities at genomic resolution – at least as efficient as OTU data and more efficient than imagery – will greatly facilitate future ecological surveys and open the door to novel ocean monitoring that is becoming more and more accessible due to improvements in DNA sequencing technology. Indeed, k-mers provide unequivocal information on the occupancy of specific genomes, as they are tracers made of fragments of organismal genomic DNA.

2. The calculation of T_min_ based on the surface layer

In this work, our goal was to analyze the contribution of surface horizontal currents to plankton biogeography. Other types of water movement certainly exist, and impact plankton biogeography. In our analyses, we do not perform a comparison of types of water movement. Instead, we focus on surface currents. We show throughout our manuscript that two-dimensional surface currents already have enough information to highlight surface plankton dynamics and that surface currents modulate, in a specific way, organismal distribution and, therefore, plankton biogeography. We believe that the relative contributions of other types of water movement (which, together with biotic responses, are very likely the source of the variations not explained by surface currents), is an appropriate and interesting topic for future studies.

3. The spatial and temporal distribution of *Tara* Oceans samples

We acknowledge that the geographical sampling of our data set, although of a global scale, is not exhaustive, as a consequence of the current limitations on genomic sequencing and practical (i.e., time) limitations on collecting samples with a sailing boat. Oceanic areas such as coastal regions and closed seas were not considered.

However, there is nothing particular about the transport in the regions that we

sampled that should prevent us from extrapolating our conclusions to other pelagic regions. It is worth noting that the link between dissimilarity and T_min_ holds true independent of the starting point, suggesting the potential existence of a general effect on community composition transported by currents within typical basin time scales. We supported this with a local demonstration of our global conclusions by focusing on the North Atlantic subtropical gyre, where sampling is very close to the Gulf Stream and where the correlation values between dissimilarity and T_min_ ranged between 0.44 and 0.86 (Figure 2—figure supplement 2). Furthermore, we performed several tests to address the effects of seasonality of our samples (described in more detail in our response below).

In our work, we tested 2 clear hypothesis and explained the results of these tests as follows: (1) the combination of large-scale spatio-temporal sampling with metagenomics permits the study of biogeographical patterns at basin and sub-basin scales, because these patterns overlay fluctuations at smaller scales, and the nature of genomic DNA makes it an excellent tracer of community composition, which requires zero *a priori* knowledge; (2) T_min_ (of horizontal surface currents) explains a significant part of plankton biogeographical variation (and is superior to the shortest geographical distance without crossing land).

For all three points above, even if our measurements of both metagenomic dissimilarity and T_min_ are the result of substantial approximations, we still find strong correlations between the two, with global values not previously reached in the literature (see lines 211-216 of the submitted version of the manuscript, unchanged in lines 224-228 of the revised manuscript). Thus, instead, we propose that our results should be interpreted in the converse manner: despite all of these potential approximations and/or biases, we found a strong, statistically significant relationship, which means that these approximations/biases are likely to be second-order in terms of explaining pelagic plankton dynamics.

We revised our manuscript to emphasize these ideas, and to respond in detail to the reviewers’ suggestions (as described in our point-by-point response below).

Finally, two general notes on our manuscript: first, due to an unfortunate find/replace error when reformatting our bioRxiv manuscript for submission to *eLife*, the titles of the 10 figure supplements were not updated in the legends associated with each figure. We have corrected this error in the revised version of our manuscript. Second, we discovered a versioning issue that affected some data matrices used in our analysis. We made minor changes to Figure 2a, Figure 3, Figure 2—figure supplement 1 and Figure 3—figure supplement 1 to correct this issue by updating them to the most recent version of the data matrices (available via our FigShare repository, DOI: 10.6084/m9.figshare.11303177). These updates did not result in any substantial changes to figure interpretation or conclusions.

Reviewer #1:1. One strong aspect of this study is that it compares community dissimilarities with travel times computed from a global ocean circulation model, which arguably makes more sense than simply considering geodesic distances between locations (as most previous studies have done). The data included with this manuscript will also be of great value to other research groups.

We appreciate the reviewer’s positive comments on our approach and on our data set.

2. The authors have shown that travel time correlates strongly with metagenomic community dissimilarities (lines 208-209), but they also found strong correlations between travel time and environmental differences (lines 232-233), and correlations between metagenomic differences and environmental differences (Figure 3 and lines 191-192). They also found that their genomic provinces often matched environmental differences. This begs the question to what extent the genomic provinces are caused by circulation within the provinces (which probably "homogenizes" communities within a province) rather than simply environmental differences between provinces (or environmental homogeneity within provinces). In other words, even if there was no circulation (thus communities are entirely determined by local environmental conditions), would we still expect to see the genomic provinces that the authors found? I would have expected a strong discussion on disentangling environmental selection effects from dispersal effects. The authors do marginally touch upon this issue by calculating partial correlations (Supplemental Figure S9), but they don't discuss this at all in the main text. Their Supplemental Figure S9c actually suggests that correlations between metagenomic dissimilarities and environmental differences (temperature and/or nutrients), when controlling for Tmin, are often stronger than correlations between metagenomic dissimilarities and Tmin (when controlling for temperature and nutrients). This might suggest that for some size classes genomic provinces are mostly caused by environmental homogeneity within basins/provinces, rather than the homogenizing action of currents. As it stands, I find the take-home messages of the paper rather vague and inconclusive from a mechanistic (i.e., rather than descriptive) point of view.

The reviewer makes an important point, and we apologize for not being clearer in the manuscript. Our results indeed highlight a strongly correlated T_min_/genetic dissimilarity/environmental conditions system which could invite the use of statistical methods for studying the independent effects of environmental conditions and circulation. However, as we explain in our manuscript and below, most or even all environmental parameters are not independent from currents; together, they are interlinked parts of the seascape. In our manuscript, we introduce the idea of using T_min_ as a framework for interpreting plankton dynamics (that is, a given window of time to study effects of physical and biological processes in plankton community composition), rather than as a physical variable itself. As noted by the reviewer, we included partial correlations in a supplementary figure in order to demonstrate that they do not affect the maximum correlation of community dissimilarity with T_min_ for T_min_ < ~1.5 years. However, we believe that, beyond this, using the results of partial correlations to draw broader conclusions from our data would be inappropriate, for three main reasons:

1. When dealing with strongly non-linear relationships, partial correlation can easily lead to interpretation problems (Vargha *et al.*, 2013 DOI 10.1007/s11135-012-9727-y). Temperature, to cite one example, has a known non-linear relationship with plankton growth (Eppley, “Temperature and phytoplankton growth in the sea”, 1972, Fish. Bull. 70(4), pp. 1063-1085). Most importantly, in a strongly correlated T_min_/genetic dissimilarity/environmental conditions system where all parameters are equivalently correlated to each other, it appears very difficult to infer who is the egg and who is the chicken given the current limitations of our data set.

2. Homogenization is not the only effect induced by currents. At mesoscale, currents also induce shearing of communities (i.e. heterogenization) just like when stirring a blob of honey in yogurt. Honey will first be stretched into filaments (heterogenization) before being diffused out on a longer timescale (homogenization). In our data set, it is impossible to disentangle the effect of stirring from the effect of diffusion on the scale of plankton communities, as it would have required us to sample all oceans at all times.

3. Finally, it would be overly reductive to interpret T_min_ as simply a proxy for the physical effect of currents on plankton distribution/displacement. Indeed, the primary components of the local environment are also transported together with plankton communities (nutrients, heat), as described at the beginning of the submitted version of our paper (lines 111-120, and references therein), and as we demonstrated in our data (Figure 3, purple line, where the correlation of nutrient concentrations and T_min_ also shows a peak at T_min_ < 1.5 years). In this framework, T_min_ therefore captures both the variation in environmental conditions and the variation in plankton communities due

to changes in environmental conditions. Hence, T_min_ does not isolate the physical effects of currents on plankton communities but instead represents the appropriate framework to capture plankton dynamics (emerging from environmental conditions/biotic interactions/effects of currents) during travel in the oceans (see also: Lévy *et al.*, 2014 DOI: 10.1215/21573689-2768549 and Dutkiewicz *et al.*, 2020 DOI: 10.5194/bg-17-609-2020 on the stretching of plankton niches by transport).

An important consequence of T_min_ being a framework for plankton dynamics rather than a proxy for physical effects is that even if we start with the hypothesis that stirring and dispersal have no effect on plankton composition, T_min_ would still be correlated with plankton dissimilarities and environmental conditions, since the simple role of conveyor of currents would still have a major impact on plankton biogeography, as they transport both plankton and nutrients and hence stretch the regions where a given plankton community would live. Without transport, plankton biogeography would certainly be different as the timescale of diffusivity is much longer than that of currents, and major paths such as the Gulf Stream in the Atlantic (see Figure 2b) would disappear.

In other words, while not being able to statistically disentangle all biophysicochemical effects on plankton dissimilarities, we nonetheless present results that provide some hints about the mechanism of the causal relationships in the T_min_/genetic dissimilarity/environmental conditions system. First, cumulative correlations clearly detect at genomic resolution plankton dynamics and environmental conditions dynamics, and second, the PCoA-RGB maps of nutrients and temperatures show patterns that are clearly similar to those of plankton communities, with differences according to size fraction (see Figure 1—figure supplement 4). While not a formal disentanglement, such results suggest that the main effect of currents is that of a conveyor transporting the plankton/nutrients ecosystem, which experiences variations in temperature as it travels, or at least that effects of diffusion and stirring on plankton and nutrients are equivalent and therefore can’t be isolated.

We thank the reviewer for pointing out that a distinct statement of the mechanism for our hypothesis was previously missing from the main text. To address this issue, we modified the text (lines 285-298 and lines 338-348) to clarify the mechanistic basis for the hypothesis we proposed to explain plankton biogeography, as follows:

Lines 285-298: “We present the following hypothesis as a potential mechanism for the partitioning of the global ocean into genomic provinces. The relatively large separation (in terms of transport time and season) among sampling stations allowed us to detect the large-scale effects of ocean circulation, which are superimposed on smaller-scale effects such as local patchiness and seasonality (as previously observed (Longhurst, 2006; Reygondeau et al., 2013)). Within ocean basins, as the intertwined dynamics of plankton and chemistry continuously occur along transport, smooth variations have emerged due to the periodic recirculation of within-basin currents. This leads to stable, continuous patterns of changes in community structure and nutrient concentrations, and also explains how temporally stable genomic provinces can exist in the face of ocean currents. Among ocean basins, depending on the sensitivity to the environment of a plankton community, higher heterogeneity in environmental conditions across different circulation patterns can disrupt the equilibrium of seascape processes within a given continuum, leading to a global delimitation into distinguishable ecological continua among different large-scale current systems, resulting in the genomic provinces that we detected.”

Lines 338-348: “Using analysis of standardized metagenomic data together with environmental and physical data, we reveal that, in a background of significant local patchiness, the integration of seascape physical, chemical and biological processes over time and space produces a quasi-stationary biological partitioning of the oceans that supersedes short-term variability and seasonal cycles, ultimately generating global biogeographical patterns. Although the strong cross-coupling among our metagenomic, environmental and physical data prevents a systematic disentangling of their various influences, we hypothesize that transport by ocean currents acts essentially as a conveyor on which interacting environmental and biological effects are layered. In this hypothesis, direct effects of currents (such as turbulent diffusivity) on plankton composition are secondary, and instead environmental and biological effects occurring during transport result in the emergence of a global plankton biogeography.”

3. The authors calculate metagenomic dissimilarities with Simka based on counts of 31bp-k-mers (instead of, say, species counts or gene ortholog counts). While this is computationally efficient, it makes a biological interpretation of their metagenomic β-diversity hard. The original Simka paper demonstrated that k-mer distances correlate with taxonomic distance matrixes, but they don't seem to examine the relationship between k-mer dissimilarities and function-centric dissimilarities, i.e., dissimilarities in metabolic functions (as inferred by functionally annotating metagenomic reads or contigs). I suspect that k-mer-based distances may not correlate well with function-based dissimilarities.

First, regarding the use of species counts, we note that we presented confirmations of all principal results of our paper using species counts (as represented by OTUs).

We agree with the reviewer that a biogeography of metabolic functions would be an interesting complement to our analyses. In fact, we attempted such a comparison in the early stages of our analysis (see below). We note, however, that it represents a different scale of evolution. Using DNA fragments (approximated by billions of 31 base pair kmers per sample) provides a very high resolution of single-base changes in plankton genomes, which are likely to be the result of individual mutations during organismal evolution. Single-base changes carry the fine-scale record of selection by environmental and biological factors at a genomic level. A gene function analysis would test different hypotheses relating to a different scale of genome evolution; that is, they would relate to the maintenance (or not) of the gene repertoire of the members of a community. In our manuscript, we show using correlation analyses that the single-base level of resolution is more appropriate than either OTUs or image-based morphology for studying the effects of circulation and environmental variation. As we wrote in our manuscript (and in the related manuscript Frémont *et al.* 2020, DOI: 10.1101/2020.10.20.347237) we believe this is because genomic polymorphism carries a more detailed record of past environmental pressure (although the distribution of this polymorphism is not necessarily the direct result of the environmental variation we observed; selection, if it exerted an influence on a given site, could have acted at another point in space or time, or on another, linked position in the genome).

Regarding the analysis of function-based dissimilarities itself, unfortunately, the vast majority of microbial plankton genomic sequence is novel, especially within eukaryotes, and is distant enough from any sequenced species that it cannot be assigned a function by sequence similarity. Therefore, we attempted this approach and we were unsuccessful, as we were unable to retrieve a sufficient proportion of annotated sequence reads (see Author response table 1). This effect was especially evident in the larger, eukaryote-enriched size fractions, likely for two reasons. First, eukaryotic genomes are generally larger and contain a substantially larger proportion of non-coding versus coding DNA (which itself could introduce a potential source of bias, as this fraction is known to vary by several orders of magnitude among plankton lineages; we note that an analysis based on randomly selected k-mers in the genome regardless of their coding potential, as we performed, should not be affected by such a bias, as we demonstrate by confirming our main conclusions using OTU and imaging data). Second, even the sequences within coding regions frequently show little similarity to any known sequence (Carradec *et al.*, 2018 DOI: 10.1038/s41467-017-02342-1).

We attempted to functionally characterize genomic provinces in the eukaryote-enriched size fractions by mapping the set of reads present in all metagenomes within the province against sequences with functional annotations from the published *Tara* Oceans metatranscriptome gene set (Carradec *et al.*, 2018) of 116 million unigenes. We obtained the results shown in Author response table 1.

**Author response table 1. sa2table1:** 

Size Fraction (µm)	0.8-5	5-20	20-180	180-2000
No. Reads Analyzed	72,431,486	43,345,056	55,994,296	71,471,864
No. Reads Matching Unigene with Functional Annotation	15,205,312	1,538,645	5,882,447	3,613,456
Proportion	0.21	0.04	0.10	0.05

We considered it inappropriate to perform functional comparisons among provinces based on such small proportions of annotations. Furthermore, there are highly uneven levels of annotation among different eukaryotic lineages, which could lead to significant biases. For example, communities in the 180-2000 µm size fraction of *Tara* Oceans samples largely consist of either collodarians or animals, with either one or the other group representing the vast majority of the community, depending on the sample (de Vargas *et al.*, 2015 DOI: 10.1126/science.1261605, Figure 5B, bottom panel). There are over 1,000 sequenced and annotated animal genomes, whereas there are zero available genomes of collodarians or closely-related species. This would result in relatively well annotated metagenomes for samples consisting mostly of animal sequences, and few or no annotations for samples consisting mostly of collodarian sequences (with the few available annotations biased towards genes with more highly conserved sequences).

For this reason, previous functional studies on marine metagenomes have been focused on prokaryotes (e.g., Ramírez-Flandes *et al.*, 2019 DOI: 10.1073/pnas.1817554116, Ulstick *et al.*, 2021 DOI: 10.1126/science.abe6301, Faure *et al.*, 2021 DOI:10.1038/s41467-021-24547-1). We anticipate that functional studies of eukaryote metagenomes will become possible in the future, as the availability of reference genomes and functional annotations increases.

Recommendations for the authorsI strongly recommend that the authors provide a quantitative and thorough discussion of my point 2 in their main text.

We appreciate the reviewer’s comments. We responded to point 2 above, and modified the main text as described in our response.

Related to point 3: Why not also perform an analysis in terms of functional (gene-centric) metagenomic dissimilarities? It seems that the authors only considered 100 million reads per sample anyway, so it should be feasible to annotate those and compare KO tables between samples. This would aid in the ecological interpretation of their findings. It would also facilitate comparison to other recent function-focused studies of marine microbial biogeography, such as [Ramírez-Flandes et al. 2019, DOI:10.1073/pnas.1817554116] or [Coles et al. 2017, DOI:10.1126/science.aan5712].

We agree with the reviewer that a functional comparison among metagenomes would also be of interest. As we explained in our response to point 3, this is not feasible given the current state of knowledge of eukaryotic genomes, compounded by the differences in current knowledge among different organismal sizes and classes (which could introduce a bias that our function-agnostic analyses were designed to avoid). We added a phrase in the conclusion of our paper (lines 351-354) to indicate that the reviewer’s proposal would be of interest in future studies.

Reviewer #2:The authors of the Tara Ocean consortium used this unique metagenomics data set of plankton size classes to test whether global biogeographic patterns exist and how these patterns are affected or even structured by global ocean currents. This data set of six size classes including viruses (<0.2 µm), pico- and nanoplankton (prokaryotes, protists), micro- and macroplankton (protists, metazoans) up to a size of 2 mm is really unique for such an effort. It provides a unified basis by using genomic DNA in all size fractions and thus avoids biases related to methodological differences of how the plankton communities of the different size classes were analyzed. By using station- and satellite-based metadata and correlating the metagenomic data of the stations and their dissimilarities to water mass transport derived from the MITgcm global ocean current model, the authors find global biogeographic patterns. Interestingly, these patterns of the different size classes are positively correlated to the minimum travel time of water masses up to 1.5 years, suggesting that biogeographic provinces evolve and presumably persist over this time span. Beyond, other factors become more important, such as temperature or the nutrient regime. The data also show that the scale of the biogeographic patterns is inversely related to the size classes. Further, a hierarchical cluster analysis of the metagenomics data of the different size classes yielded global genomic provinces which grouped together stations of related environmental and biogeochemical properties beyond water masses.The statistical evaluation is very rigorous and for various correlations multiple and different analyses have been performed. The pairwise dissimilarity analysis of stations was done both for the metagenomics and OTU data yielding basically similar results and thus confirming the validity of the outcome.Because of the novelty of this analytical approach this study may become seminal for further similar analyses. In order to become a really solid basis for such future analyses I suggest that the authors should consider the following critical points to further improve the study and to revise the manuscript accordingly.

We thank the reviewer for their positive comments on our results and methodology. We respond to their suggestions below.

1. I tried to find specifications for sampling of the larger plankton size classes in this manuscript and other Tara Ocean publications. According to the available information metazoans were also collected from Niskin bottles. If this is correct it means that only rather few organisms per sample and depth were collected, assuming a total abundance of <20 animals per liter in oligotrophic regions. So the questions arises how reliable the data on the larger size classes are for individual taxa compared to the smaller size classes where this point is not an issue. Usually zooplankton is collected by net hauls to obtain enough material. However, if net hauls are used they integrate over sections of the water column and it is impossible to sample a particular depth such as DCM. This point may also be an issue for the cluster analysis of the genomic provinces.

The reviewer is absolutely correct that different protocols are required to capture the diversity of smaller and larger plankton. But *Tara* Oceans sampling protocols were in fact designed to sample organisms larger than 20 μm with nets. The sampling protocols and the reasoning behind them are described in the manuscript we cited in the Methods section (lines 328-330 of our initial submission and 370-372 of the revised submission; Pesant *et al.*, 2015 DOI: 10.1038/sdata.2015.23, section 6). This is accompanied by an extensive technical validation section in the same manuscript. Unfortunately, due to the large size and scope of the *Tara* Oceans expedition, it was not possible to include a detailed description of the validation of the sampling protocols in our manuscript.

Specific protocols were in place to address multiple considerations related to the use of plankton nets: sampling depth, method of towing the net (oblique/vertical/horizontal), speed of towing the net, and time of day (i.e., day vs. night). For the net sampling of the larger size fractions of plankton communities analyzed in our manuscript, “Nets were lowered to the selected environmental feature and towed horizontally for 5-15 min at a speed of 0.3 m/s” (Pesant *et al.*, 2015). Environmental features included both the surface and DCM (described in section 5 of the paper). Thus, every effort was made to ensure an appropriate sampling of plankton of different sizes at particular depths.

Therefore, while we understand the concerns of the reviewer regarding possible small volumes of water sampled for our larger size fractions of plankton and regarding potential sampling bias for DCM, these were not the case (importantly, we also note that most analyses and conclusions presented in our paper concern surface samples only).

2. The cluster analysis of the genomic provinces shows that at quite a few stations the sample near the surface and at the DCM affiliates to different provinces. For the analysis of the water transport the surface layer of the MITgcm model was used. However it is known that current velocity and direction near the surface and at lower depths of the mixed layer may diverge, in particular towards the equatorial currents (Cravatte et al. 2017, J. Phys. Oceanogr. 47: 2305, DOI: 10.1175/JPO-D-17-0043.1; Hu et al. 2020, Sci. Adv. : eaax7727, DOI: 10.1126/sciadv.aax7727). How would the water transport and plankton dispersal change if only the near surface samples were used for this analysis (which would be the correct way for this analysis) or if this analysis were done with the MITgcm model for the depth section of the DCM (which would imply a reduced number of stations and an adjustment of depths because the depth of the DCM was variable)?

The reviewer makes two suggestions. First, the reviewer suggests that we restrict our correlation analyses to surface samples. In fact, this is what we did, as described in the Methods (lines 516-519 of our initial submission): “We calculated rank-based Spearman correlations between β-diversity, T_min_ and environmental parameters (…) for surface samples with a Mantel test with 1,000 permutations and a nominal significance threshold of p < 0.01.” We clarified this statement by modifying this section of the Methods in our revised manuscript (lines 555-556), which now begins: “Our correlation analyses were restricted to *Tara* Oceans samples collected at the surface, and did not include DCM samples.”

Second, the reviewer suggests that we compare a simulation of travel times at the depth of the DCM to the samples collected at the DCM. Unfortunately, due to computational limitations, even given the significant resources available to our colleagues at MIT who performed the simulation, it was not possible to produce a simulation using the MITgcm with a variable depth matching the depth of the DCM at each simulated position. However, our colleagues did produce a simulation at a fixed depth of 75 meters, well below the impact of the wind (i.e., below the core of the Ekman layer in most cases; see also the introductory comments above in our response). We did not present analyses of the 75 meter simulation in our initial submission due to space limitations. The simulation depth is roughly equivalent to the median DCM depth of 58 meters in the samples we analyzed (calculated from depth data available in Supplementary Table 2). To address the reviewer’s concern, we performed a correlation analysis of β-diversity for DCM samples and T_min_ from the 75 meter depth simulation. Author response table 2 shows Spearman correlation values and the number of stations in each calculation, in comparison to the original surface/surface correlations:

**Author response table 2. sa2table2:** 

Size Fraction (µm)	Surface	DCM
	Correlation	No. Stations	Correlation	No. Stations
0-0.2	0.55	40	0.63	32
0.22-1.6/3	0.44	62	0.40	45
0.8-5	0.46	77	0.50	53
5-20	0.33	50	0.43	31
20-180	0.38	80	0.38	44
180-2000	0.45	80	0.43	54

It can be seen that, globally, the correlation levels we observe between DCM samples and the DCM simulation are comparable to those between surface samples and the surface simulation. As correctly noted by the reviewer, there are fewer DCM samples available for comparison. This relative lack of samples prevents us from presenting a cumulative correlation analysis for the DCM (as in Figure 2a for surface samples), since the paucity of samples in ranges of T_min_ resulted in correlations which were not significant at our nominal threshold of p < 0.01.

3. Stations in oceanic gyres dominate the sampling grid of the Tara Ocean expedition and stations in coastal and equatorial upwelling regions are greatly underrepresented. Therefore, and based on some discussion in the manuscript, the impression emerges that the biogeographic patterns and their relationship to Tmin is mainly true for oceanic gyres. I suggest that the authors should elaborate on this point and may also consider these constraints in their biogeographic analysis.

The reviewer is correct to note that the relationship we describe between β-diversity and T_min_ is true for open ocean samples, and not for coastal/upwelling samples. In fact, our correlation analyses explicitly excluded samples outside the open ocean, as we described in the Methods section (lines 512-513 of our initial submission, lines 556-558 of the revised submission): “We excluded stations that were not from open ocean locations from correlation analyses to avoid sites impacted by coastal processes.”

Our biogeographic analyses included all samples, including coastal samples. We discussed the relationship between upwelling zones and genomic provinces inSupplementary Information 4 (lines 955-985 of our initial submission, lines 1015-1060 of our revised submission).

In order to clarify the text in response to the comment raised by the reviewer, we revised the main text in the following locations:

Lines 129-134: “Our sampling largely focused on open ocean sites located in the main gyres, but also included other areas with distinct oceanographic features, such as coastal upwelling zones and lagoons (Figure 1—figure supplement 1b). We chose to study biogeographic patterns along large-scale currents in the principal oceanic gyres, with counterpoints to other oceanographic features in which the influence of ocean transport by the main currents is likely to be relatively weaker, such as upwellings.”

Lines 183-186: “Although the large majority of our samples were located in oceanic gyres, samples located in physically distant zones but with shared environmental conditions, such as oceanic upwellings, also grouped together (for example, genomic province B6 in the bacterial-enriched size fraction).”

Lines 337-338: “In this study, we provide genomic evidence for an organism size-dependent global-scale open ocean plankton biogeography shaped by currents.”

4. An important outcome of the study are the different scales of the dispersal of the size classes with Tmin. In a previous publication of the Tara Ocean consortium (Sunagawa et al. 2015, Figure 4B) they show a plot of an increasing dissimilarity of the prokaryotic communities with distance up to appr. 5000 km. In this manuscript the authors mention that they calculated correlations of the dissimilarities with distance for the different size classes but do not show any data. The study would greatly benefit when they show these plots for the different size classes which should yield different patterns. The distance of 5000 km may relate to the travel time of 1.5 years over an oceanic gyre.

We agree with the reviewer that it is important to show the relationship between β-diversity and geographic distance. In fact, we presented this analysis in our initial submission (lines 218-220; lines 232-234 of the revised submission): “In contrast, no such unimodal pattern was found for correlations between metagenomic dissimilarity and geographic distance (without traversing land; Figure 3—figure supplement 1f).” The accompanying figure shows the cumulative correlation of geographic distance and β-diversity, as measured by metagenomic dissimilarity for all 6 organismal size fractions, and estimated from imaging data.

The distance of 5,000 kilometers is indeed on the order of the distance to cross an oceanic gyre. Nonetheless, Figure 3—figure supplement 1f shows that geographic distance is not the appropriate framework for studying differences in plankton genomic community composition. Instead, our results demonstrate that travel time is the appropriate framework to use. To respond to the reviewer’s comment, we modified the main text as follows (lines 251-252): “Up to ~1.5 years of travel time, the timescale of large-scale transport is therefore the appropriate framework for studying differences in plankton genomic community composition (Figure 2b).”

In addition further recommendations are as follows:l. 180-183, 195-199 and other places: There are quite a few genomic provinces of the prokaryotic and protest enriched size classes which go far beyond one ocean basin and even occur at one station near the surface and at the DCM. So be more precise in describing these features. Howe would you cope with genomic provinces which encompass similar stations but in corresponding gyres of the northern and southern hemispheres?

This is an astute observation on the part of the reviewer. In our opinion, this demonstrates that the genomic provinces that we defined based on pairwise similarity are not limited to samples collected at geographically close stations. Instead, this shows that the data we analyze represent community similarities at a spatio-temporal scale overlying the influences on variation at more local scales. This is complementary with the observation that upwelling stations are located in the same genomic provinces, despite being separated by very large geographic distances (in this case, the role of surface connectivity is eclipsed by strong variation in local environmental conditions).

In the main text of the paper, we focus mainly on biogeographical patterns for surface samples and their comparison to travel time simulated at the surface. In order to avoid confusion, we discuss the biogeography of DCM samples in the supplementary information. We modified the supplementary information to respond to the reviewer’s comment regarding stations whose surface and DCM samples were in the same genomic province, in Supplementary Information 3 (lines 1006-1013), as follows: “Although our analyses of the influence of transport and other environmental conditions on plankton biogeography are focused on surface samples (the depth at which our simulations of transport were conducted), we included samples collected at the deep chlorophyll maximum (DCM) in our genomic provinces in order to produce a broader description of plankton biogeography. In general, samples at the surface and DCM from the same station clustered together in the same genomic provinces, although we observed a minority of cases in which surface and DCM samples from the same station were found in different genomic provinces, and even a few genomic provinces composed only of DCM samples (e.g., C6 in the 0.8-5 µm size fraction; Figure 1—figure supplement 4).”

To further address the reviewer’s point, we modified lines 183-186 of the text in order to be more precise in describing genomic provinces, as follows: “Although the large majority of our samples were located in oceanic gyres, samples located in physically distant zones but with shared environmental conditions, such as oceanic upwellings, also grouped together (for example, genomic province B6 in the bacterial-enriched size fraction).”

l. 181: delete "to" in this part of the sentence:.….tended to be limited to a single ocean basin and [TO] approximately correspond to…

In this sentence, we chose to employ “to” as a syntactic marker of the infinitive in the clause that follows the coordinating conjunction (“and”). The reviewer suggests using the bare infinitive. Both forms are grammatically correct; we prefer to retain the full infinitive as we believe it conveys our intended meaning more precisely.

l. 366: must be "….the same number of reads WAS used…."

We thank the reviewer for this correction, which we implemented as suggested (line 409).

l. 1019-1024: You hypothesize that the travel time of 1.5 years is equivalent to the time needed for crossing an oceanic gyre. I assume that this must not remain a hypothesis because I am convinced that ground truth data exist which provide such travel times, may be from the Argo floats program.

We apologize for the confusion due to the wording of this paragraph. In agreement with the reviewer’s statement, in our hypothesis, we take the average crossing time of an ocean gyre as an assumption. However, the hypothesis does not depend on the exact crossing time of an ocean gyre, *per se*. To clarify this point, we have reorganized this paragraph to remove direct references to a fixed travel time (lines 285-298).

The time estimate to cross the basins is based on assuming an average current velocity of 20 cm/s, which are the typical values for the mean surface currents (e.g., Lumpkin and Johnson, 2013 DOI: 10.1002/jgrc.20210). This gives a range of 6,000 km for the typical drift length, which compares with the longitudinal size of the N. Atlantic subtropical gyre (about 5,000 km). Notably, for the N. Atlantic subtropical gyre travel time, estimates from previous work on the European eel (based on otoliths and on high resolution Lagrangian computations) confirm our estimates, suggesting additionally that the eastward journey could be actually even faster (about 10 months) (Bonhommeau *et al.*, 2009 DOI:10.1111/j.1365-2419.2009.00517.x, Rodríguez-Díaz and Gómez-Gesteira, 2017 DOI:10.1016/j.seares.2017.06.010).

Reviewer #3:This is another contribution from the Tara consortium and collaborators, in this case, they try to correlate ocean circulation with plankton biogeography. They have done a number of statistical comparisons using metagenomic data to analyze the effect of a parameter that they call Tmin, the minimal travel time, an estimation of the time that would transfer a water volume from one station to another as deduced from the expected water-mass movement. The problem of this approach (as with previous Tara papers in the view of this reviewer) is the random distribution of Tara stations at similar latitudes and (largely) from the surface and at different seasons. They contemplate the ocean as a two-dimensional system, largely ignoring the third dimension (depth) and the water column stratification that appears at most of these stations seasonally. Surface water movements have been considered without regard to the potentially more important vertical ones that happen when the water column mixes in colder seasons. Thus, their conclusions are flawed by a poor sampling strategy. The authors could have used more structured sampling efforts such as Geotraces, at least to check their overarching conclusions.

The reviewer raises 4 related concerns, which we address here. First, in our manuscript, we do not claim that horizontal transport explains all of plankton biogeography, nor that vertical transport does not have an influence. Instead, we find strong and statistically significant correlations between surface plankton community dissimilarity and travel times from simulations of horizontal transport at the surface (discussed in more detail in our responses to Reviewer #1). These correlation values surpass those from previous global-scale studies (see lines 211-216 of the submitted version of the manuscript, unchanged in lines 224-228 of the revised version). To us, this indicates that the analysis of transport by horizontal ocean currents can make a valuable contribution to understanding the forces that shape plankton community structure. It does not mean that other forces, such as vertical transport, do not also play important roles. We hope that our study will serve as a basis for future work that explores these roles (although we are unaware of any currently existing data set or study on the effect of three-dimensional water movement on plankton community composition at a global scale).

Second, we tested for the effects of seasonality (using daily sunshine duration as a proxy) by asking whether it was significantly different among genomic provinces, as was the case for temperature and various nutrients. These analyses were presented in Figure 1—figure supplement 7 (panels a and b) of the originally submitted version of our manuscript (unchanged in the revised version). Sunshine duration was significantly different among genomic provinces in only 1 of 6 organismal size fractions (180-2000 µm; panel a). Moreover, there were no significant differences when focusing on temperate latitudes, by removing stations near the Antarctic (panel b). Therefore, we concluded that the effect of seasonality was not as pronounced as the effects of temperature and nutrients.

Third, as we described in our introductory remarks above, and in lines 223-231 of the submitted version of our paper (unchanged in lines 252-261 of the revised version), we view the potential limitations of the sampling strategy instead as a strength. Despite the fact that the samples were collected across four different seasons and three different years, we nonetheless find strong and statistically significant correlations between the variation of plankton community composition of those samples and simulated minimum travel time (as well as variation in environmental conditions). We believe this places a lower bound on the strength of the effect that we observed.

Fourth, GEOTRACES (Biller *et al.*, 2018 DOI: 10.1038/sdata.2018.176) and more recent efforts such as Bio-GO-SHIP (Larkin *et al.*, 2021 DOI: 10.1038/s41597-021-00889-9) are inappropriate data sets for our analyses for two reasons. The first is biological and the second is technical: (1) These projects did not separate organisms by size fraction prior to metagenomic sequencing. This confounds multiple organismal size fractions (viruses, bacteria, microbial eukaryotes and planktonic animals). Size is the dominant factor differentiating plankton communities in the global ocean (de Vargas *et al.*, 2015 DOI: 10.1126/science.1261605, Figure 6A). In addition, the major part of the biomass of the ocean is divided roughly evenly among bacteria, protists and animals (Bar-On and Milo, 2019 DOI: 10.1016/j.cell.2019.11.018), meaning that a sampling strategy that does not separate organisms by size is likely to collect a mixture of these three groups (and a relative paucity of viruses). To avoid these issues, we instead analyzed samples separated into 6 organismal size fractions, each of which we studied independently, such that our genomic analyses are carried out among organisms of a similar size. This avoids comparing the genomes of completely different organisms (e.g., bacteria vs. animals), while at the same time permitting us to produce insights on differences among plankton of different size classes (see, for example, Figure 2a, Figure 1—figure supplement 4a-f, Figure 1—figure supplement 7, Figure 2—figure supplement 2 and Figure 3—figure supplement 1b). (2) Both studies lack the necessary sequencing depth for our calculations. GEOTRACES has a mean of 27 million sequenced paired-end reads per sample (Table 4 of Biller *et al.*), and Bio-GO-SHIP has roughly 26 million (derived from Table 1 of Larkin *et al.*). We found that, for the five non-viral enriched size fractions in our study, it was necessary to compare at least 100 million reads before the results of our analyses stabilized (see Methods, “Calculation of metagenomic community dissimilarity”). The relative lack of sequencing depth of both GEOTRACES and Bio-GO-SHIP would be expected to be compounded by the fact that multiple organismal size classes are mixed together in the same sample, thus further reducing the proportion of each size class sampled. For these two reasons, we do not believe it would be appropriate to perform a comparison to GEOTRACES or Bio-GO-SHIP.

A second major flaw of this work is that the main source of information is what they call "metagenomic dissimilarity". It is actually the reciprocal of the ratio of shared (100% identity) 31 nucleotide K-mers between stations out of a pool of several million Illumina reads. This is a quite rough estimate of similarity that does not contemplate the nuances of genome features from virus to zooplankton. For example, the presence of IS or related elements in prokaryotic genomes might bias this parameter strongly, as would the presence of multiple repeats of rRNA genes in eukaryotic cells. The biological significance of metagenomic dissimilarity should be carefully assessed. I do not imply that it cannot be used, but to reach conclusions of such weight ("oceanic genomic provinces") a much more refined sampling strategy and analysis of the data would have been required. For example, why were the myriad of MAGs derived from prokaryotes and viruses at different geographical sites not considered? At least as a control for their claims. Actually, the several reports of nearly identical genomes at different oceanic provinces points towards the opposite. I do not believe the evidence presented here warrants the kind of conclusive statements presented.

We note that all the principal results we obtained with metagenomic dissimilarity were confirmed with OTU-based data, which cannot be affected in the same way by the biases raised by the reviewer. These confirmations were presented in Figure 1—figure supplement 2, Figure 1—figure supplement 5, Figure 2—figure supplement 1, and Figure 3—figure supplement 1 and discussed in Supplementary Information 1, all in the submitted version of our manuscript (and which remain in the current version).

In addition, we did, in fact, confirm our results with MAG-based data (lines 297-303 of the submitted version of our paper, and figures referenced therein; unchanged in lines 243-249 of the revised version), and also with imaging-based data (lines 287-291 of the submitted version of the paper, and figures referenced therein; lines 235-243 of the revised version).

More importantly, the potential confounding factors in genome sequences proposed by the reviewer (prokaryotic IS elements and intragenomic repeats of eukaryotic ribosomal genes) would not be expected to strongly bias our results. We will respond regarding the ribosomal RNA locus in eukaryotes, as eukaryotes dominate the majority of the size fractions we analyzed (4 of 6); any strong bias present in 4 of 6 analyzed size fractions would be clearly evident in our data.

A typical eukaryotic ribosomal RNA locus is 5 kb (5 x 10 ^3^), but eukaryotic genomes can be up to 6 orders of magnitude larger (gigabases, 10^9^), or more. Therefore, even thousands of rRNA repeats will have no substantial impact on genome-wide comparisons of randomly-sampled sequences.

How many rRNA repeats are expected to be present in the genomes we sampled? Nearly all genomes of planktonic microbial species have been assembled with Sanger or Illumina short read sequencing, which result in difficulties assembling repeated ribosomal RNA loci (and thus in counting their number of copies in a genome). A recent study on the diatoms *Phaeodactylum tricornutum* and *Thalassiosira pseudonana*, which was among of the first to use long reads for microbial plankton genome assemblies, provides an opportunity to assess the contribution of rRNA copies to total genome size (Filloramo *et al.*, 2021 DOI:10.1186/s12864-021-07666-3). This study found 7 copies of the rRNA locus in *P. tricornutum* of average length 5,936 bp, and 5 copies in *T. pseudonana* of average length 5,827 bp. The genomes of both species are estimated to be roughly 30 Mb in size, meaning that rRNA locus copies represent approximately 0.001 of each genome. This proportion would thus have a negligible (≤ 0.001) effect on genome-wide comparisons of matching k-mers such as those we performed, as k-mers are derived from sequencing reads randomly sampled from community genomic content.

In the absence of completely finished genome sequences, a method to estimate 18S gene copy number from short read mapping counts has also been proposed (Gong and Marchetti, DOI: 10.3389/fmars.2019.00219). This method estimated 18S copy numbers ranging from ~3 to ~161 for 7 different species of microbial eukaryotic plankton. The only two of these species with 18S copy numbers on the order of 10^2^ were *Symbiodinium kawagutii* and *Symbiodinium minutum*; members of the genus *Symbiodinium* are estimated to have genome sizes on the order of 10^9^ (LaJeunesse *et al.*, 2005 DOI:10.1111/j.1529-8817.2005.00111.x). Thus, consistent with the results described above, even the highest of these estimates would have a negligible effect on our comparisons.

Finally, intragenomic variation at the ribosomal RNA locus has been observed across the eukaryotic tree of life: in dinoflagellates (Thornhill *et al.*, 2007 DOI:10.1111/j.1365-294X.2007.03576.x), apicomplexans (Li *et al.*, 1997 DOI:10.1006/jmbi.1997.1038), cercozoans (Bass *et al.*, 2012 DOI:10.1371/journal.pone.0049090), foraminiferans (Weber and Pawlowski, 2014 DOI:10.1016/j.protis.2014.07.006), animals (Gasmi *et al.*, 2014 DOI:10.1186/s12983-014-0084-7), amoebozoans (Kudryavtsev and Gladkikh, 2017 DOI:10.1016/j.ejop.2017.09.003), diplonemids (Mukherjee *et al.*, 2020 DOI:10.1111/1462-2920.15209) and rhizarians (Sandin *et al.*, 2021 DOI:10.1101/2021.10.05.463214).

Thus, even if rRNA repeats were present as a substantial fraction of the genome (which has not been observed in any sequenced eukaryotic genome to date), it is likely that our approach, which is sensitive to changes at a single-base resolution (i.e., it relies on exact matches among 31 nucleotide k-mers), would still be able to detect variation among different copies of the locus.

In terms of other repeats, such as regions of low complexity resulting from simple repeated sequences, the software we used to produce dissimilarity matrices, Simka, includes an option to filter for such regions (Benoit *et al.*, 2016 DOI: 10.7717/peerj-cs.94). We added a phrase to the methods (lines 403-404) to clarify that we used this feature in our analyses, as follows: “Within Simka, we filtered regions of low complexity with -read-shannon-index set to 1.5.”

In what follows I have identified specific points that would need clarification or modification in case the work had to be published.Ln 98. There are now many studies on the biogeography of microbes based on metagenomics, including depth profiles and similarities along different transects so this sentence is just wrong.

Although we agree that there are now many studies on the biogeography of microbes based on metagenomics, including studies we ourselves published and referenced in our manuscript (e.g. Vannier *et al.*, 2016, DOI: 10.1038/srep37900), we believe that it nevertheless remains true that biogeographical studies have traditionally focused on readily visible organisms. For example, as stated in the first paragraph of the review article we referenced (“Microbial biogeography: putting microorganisms on the map”, Hughes Martiny *et al.*, 2006, DOI: 10.1038/nrmicro1341), “Since the eighteenth century, biologists have investigated the geographic distribution of plant and animal diversity. More recently, the geographic distributions of microorganisms have been examined.”

Ln 102 seascape= metadata

We are not sure that we understand the comment by the reviewer. The seascape is not equivalent to metadata, analogous to how the landscape is not equivalent to metadata in terrestrial ecosystems. Further details on the seascape are available in the book we cited entitled “Seascape Ecology” (Pittman SJ (ed.), 2017, Wiley-Blackwell, 526 pp). Our use of the seascape was described in more detail in the submitted version of our manuscript (lines 118-120 and preceding text, unchanged in the revised version).

Ln 104 the approach is not consistent (e.g., amplicon sequencing and metagenome similarity)

This sentence describes our metagenomic analyses, since it follows the previous sentence, which explicitly refers to metagenomes, and does not mention OTU data: “Here we assessed the global structure of plankton biogeography … by analyzing metagenomes of plankton communities” (lines 100-104, submitted version, unchanged in revised version). Furthermore, metagenomic analyses are the principal focus of the main text of the manuscript, and thus, we believe, are most appropriate to describe in the abstract. See lines 159-160 of the submitted version of our manuscript (unchanged in lines 163-164 of the revised version): “We focus on analyses of metagenomic dissimilarity here, with accompanying results for OTU dissimilarity presented in Supplementary Figures.”

We obtained metagenomic dissimilarities and OTU dissimilarities for each size fraction. Therefore, our approach is consistent within the analysis of metagenomic dissimilarities, and, in parallel, consistent within the analysis of OTU dissimilarities.

Ln112 and mixing with deeper layers

The use of the term “mixing” implies a process that can occur both horizontally and vertically. We believe it would be superfluous to specify both of these possibilities here.

Ln118 neighboring (and deeper) water masses

The ocean is three-dimensional. Neighboring water masses can be located at equal depths, higher depths, and lower depths.

While this is true, we acknowledge that the ocean simulations used in our analyses include only horizontal transport. Thus, to avoid ambiguity in the interpretation of our results, we do not explicitly refer to other types of transport and mixing in our introduction.

Ln 155 However, some taxa information would have enriched enormously the manuscript

We believe that an in-depth exploration of the taxonomic composition of these samples merits a description in an independent manuscript. This companion manuscript has now been published (Sommeria-Klein *et al.*, 2021 DOI: 10.1126/science.abb3717).

Ln 160 MAG abundance is not a reliable estimate of microbe abundance, often it is the opposite i.e. assembled microbes are not particularly abundant in the environment as exemplified by SAR11 or picocyanobacterial (several references).

SAR11 (and to a lesser extent *Prochlorococcus* and *Synechococcus*) is a very unusual example, where a large number of abundant and closely related populations coexist, preventing optimal assembly and binning (Nayfach *et al.*, 2016 DOI:10.1101/gr.201863.115, Tully *et al.*, 2018 DOI: 10.1038/sdata.2017.203, Delmont *et al.*, 2018 DOI: 10.1038/s41564-018-0176-9, Delmont *et al.*, 2019 DOI:10.7554/*eLife*.46497.001). In contrast, we could list more than 100 lineages for which genome-resolved metagenomics does recover the most abundant genomes within the plankton (Delmont *et al.*, 2020, DOI: 10.1101/2020.10.15.341214). Within the eukaryotes, for instance, Delmont *et al.* recover the most abundant genomes of prasinophyceae (*Bathycoccus*, *Micromonas*, etc.), as well as 34 diatom and 215 copepod MAGs. As a result, especially in large size fractions, the MAG database that we used recruits a substantial portion of reads (for example, an average of approximately 20% in the 20-180 µm size fraction; Figure 1 in Delmont *et al.*, 2020), with lineages that, once again, do not have the same population characteristics as compared to SAR11.

So, although we agree that a relatively small set of MAGs cannot alone explain all the complexity of plankton, we nonetheless value the new opportunities the Delmont *et al.* database offers in terms of understanding the distribution and phylogeny of abundant eukaryotic lineages.

Ln 163 explain "significant"

We agree with the reviewer that “significant” could be misinterpreted as referring to statistical significance. We replaced “significant” with “substantial” and added the text “average pairwise dissimilarity > 80%” as part of the parenthetical phrase referencing the figures that present these data (lines 167-169), as follows: “Globally, we observed substantial metagenomic dissimilarities (average pairwise dissimilarity > 80%) between sampled stations (including adjacent sites) across all size fractions (Figure 1—figure supplement 3a, Supplementary Information 1).”

Ln 165 to the end of the paragraph. Extremely subjective i.e., heterogeneous compared to what? A depth profile will show that microbes at a 50 m distance in depth are likely more dissimilar than those located a 500 Km but at the same depth.

The average metagenomic dissimilarity among pairs of samples, within all size fractions and across both sampling depths (surface and DCM), is greater than 80% (Figure 1—figure supplement 3). The minimum metagenomic dissimilarity among any pair of samples is greater than 50%, meaning that all pairs of samples are more different from one another than they are similar. We consider this to be objectively heterogeneous.

We note that the hierarchical clustering that we performed answers the question of how samples separated by different distances and sampled at different depths are related to one another, by grouping together samples that are more closely related to one another than they are to samples outside the group (Figure 1—figure supplement 4 and Figure 1—figure supplement 6). There are instances in which distantly-located samples from the same depth are more closely related to one another than they are to closely-located samples at different depths. As an example, in the 0.8-5 µm size fraction, across large distances in the Pacific Ocean, surface samples in the C4 genomic province are more closely related to one another than they are to DCM samples in the C7 genomic province. Conversely, there are also examples in which samples at different depths cluster geographically, such as the C9 genomic province of the 0.8-5 µm size fraction, also located in the Pacific.

Ln 175. Colors are a very subjective representation, different shades of grey or lines of different thickness connecting stations as actually presented in Figure 2 b are easier to interpret.

These colors are objectively obtained by calculation from the three axes that describe the largest amount of variation in our principal coordinates analyses within each size fraction (as explained in the Methods, lines 435-445 of the submitted version of our manuscript, unchanged in lines 478-488 of the revised manuscript). The RGB color code permits us to display three axes of variation, whereas a gray scale could not, and the resulting referential is orthogonal, meaning that one specific color can only be associated with one specific location in the eigenvector space. Using a combination of lines or other symbols would not be possible given the limits of resolution for global geographical maps.

Furthermore, no statistical analyses of differences among stations or among genomic provinces are conducted using these color data. Statistical analyses are instead conducted directly on the underlying matrices of dissimilarity. We believe the colors are helpful for the reader to gain a rapid understanding of the variation among stations (analogous to how ordination plots are a frequently used, standard method to visualize the relationships among a set of samples).

Ln 187 surface temperature?

Yes, the reviewer is correct. We thank them for this suggestion and we changed the wording to read “Seawater surface temperature” (line 194).

Ln 187 temperature or the cognate community?

We do not understand what a “cognate community” is. Seawater surface temperature was significantly different among the sets of samples defined as genomic provinces using metagenomic dissimilarity.

Ln 190 which size classes?

We tested six size fractions and 12 environmental conditions. Instead of listing in the text all 30 pairwise combinations of size fractions and environmental conditions that display significant differences among genomic provinces, we refer the reader to Figure 1—figure supplement 7, which contains a visual description of all significant differences, as well as additional analyses of these differences.

Ln 194 or those microbial communities vary more sharply with depth and when they upwell (with nutrients) disrupt the water-mass continuity more

We agree with the reviewer that upwelling, depending on its strength relative to other sources of water transport, could exert a potentially divergent effect on communities in different size classes. We mentioned the effect of upwelling zones in the main text (lines 250-252 of the originally submitted version of our paper; lines 281-284 of the revised version) and, more specifically and in more detail, the effects of upwelling in the context of organismal size in Supplementary Information 4 (lines 955-1000 of the submitted version; unchanged in lines 1015-1060 of the revised version).

In addition, to address a point raised by Reviewer #2, we modified lines 183-186 of the text to highlight the effects of upwelling, as follows: “Although the large majority of our samples were located in oceanic gyres, samples located in physically distant zones but with shared environmental conditions, such as oceanic upwellings, also grouped together (for example, genomic province B6 in the bacterial-enriched size fraction).”

Ln 197 and vertical transport?

Roles for many types of transport may be suggested by our observations on community dissimilarity. Here, we highlight the particular role of surface transport, because the following sentence introduces the comparison of metagenomic dissimilarity to simulations of surface transport. Therefore, we believe it would be superfluous to catalog other potential types of transport here.

Ln 219 In any case, it would be interesting to know how dissimilarity correlates with geographic distance as well since Tmin will vary accordingly at shorter distances. It is to be expected that a transect following the Gulf Stream (small Tmin) will show high similarity. This would be a good control of the method of metagenome comparison.

Both of these suggestions were already implemented in the submitted version of our manuscript.

We agree with the reviewer that the comparison of geographic distance (without traversing land) to metagenomic dissimilarity is of interest. This correlation was presented in the originally submitted version of our manuscript as Figure 3—figure supplement 1f, as indicated in the text (line 220 of the submitted version; unchanged and referenced in lines 233-234 of the revised version).

Concerning the suggestion about the Gulf Stream, we agree. We presented this analysis(of both metagenomic dissimilarity and OTU-based dissimilarity) in lines 256-260 of our submitted manuscript (unchanged in lines 304-308 of the revised version), accompanied by Figure 2—figure supplement 2 and Supplementary Information 6.

Ln 223 Even more important would be the season and whether the water column is stratified or mixed as would be the case in winter in temperate latitudes (most of Tara samples).

We tested for the effects of seasonality (using daily sunshine duration as a proxy) by asking whether it was significantly different among genomic provinces, as was the case for temperature and various nutrients. These analyses were presented in Figure 1—figure supplement 7 (panels a and b) of the originally submitted version of our manuscript (unchanged in the revised version). Sunshine duration was significantly different among genomic provinces in only 1 of 6 organismal size fractions (180-2000 µm; panel a). Moreover, there were no significant differences when focusing on temperate latitudes, by removing stations near the Antarctic (panel b). Therefore, we concluded that the effect of seasonality was not as pronounced as the effects of temperature and nutrients.

Ln 239 temperature is more correlated with depth and season, particularly at the temperate latitudes.

We used the term Earth-scale process to refer to the phenomenon that significant differences in temperature are driven by seasonal cycles and latitude, which vary on large scales. Depth is a ubiquitous pattern and would be better characterized by its gradient (first derivative) than the temperature *per se*. Therefore, we agree with the reviewer, and in fact this is what we meant for Earth-scale processes: change in latitude and in season.

Figure 2. Many points appear very divergent, can you explain the most extreme cases and label them in the figure?

These points may appear divergent due to the reduced limits of metagenomic similarity on the vertical axis (from zero to roughly 15%, rather than from 0-100%). Furthermore, the gray shaded area around the black line represents the 95% confidence interval of the exponential fit, and is not directly representative of the variation present in the data. Thus, points outside this area should not necessarily be considered divergent.

Nonetheless the reviewer proposes an interesting idea to explain the points that are found farthest from the line of exponential fit. In preliminary work for our paper, we found that these points often corresponded to zones with large differences in environmental conditions. Nevertheless, we chose to keep the current manuscript focused on global-scale processes. We agree with the reviewer that these specific cases could be explored in future work.

Ln 332 what happens when the water column is mixed and there is no DCM?

The reviewer raises an important point. Our genomic provinces could potentially be affected if there were a substantial proportion of stations at which the DCM could not be identified. However, this was the case for only 2 of the 124 *Tara* Oceans stations in our data set. In general, the choice of the “DCM” depth by the chief scientist on board was to either target the DCM (from fluorescence profiles) or another relevant feature, if the former was not clearly identifiable. Typically, in the latter cases, the target was the base of the mixed layer (e.g., in the Equatorial Pacific). This information is available in Supplementary Table 1 (the depth of the affected samples is labeled as “MXL”).

We emphasize that the correlations between community dissimilarity, T_min_ and environmental conditions that we calculated would be unaffected by aspects of sampling related to the DCM, as our correlation analyses concern only the ocean surface (samples collected at the surface compared to a simulation conducted at the surface).

Ln 252 easy enough to pinpoint upwelling areas

We agree with the reviewer that it would be straightforward to identify samples collected in upwelling zones, although we note that these would represent only one example of stations experiencing similar oceanographic phenomena, and thus we would not expect these stations alone to explain our observation that correlations of metagenomic dissimilarity with differences in temperature increased beyond ~1.5 years of minimum travel time. Nonetheless, we added the identity of these samples to the main text (lines 281-284), as follows “This might be related to distant Tara Oceans stations experiencing similar oceanographic phenomena (notably temperature), for example upwelling zones (stations 67, 92 and 135; Figure 1—figure supplement 1), producing generally similar environmental conditions.”

Ln 253 There seems to be something wrong with the plots presented in Supplementary Figure 2. If anything, they seem to prove that there is no clear correlation between OTUs and metagenomic dissimilarity what is actually not surprising considering the difficulty when trying to correlate data obtained in so different approaches and with different types of genomes (prokaryotic versus eukaryotic or even multicellular planktonic organisms).

The reviewer correctly notes that the vertical and horizontal axis labels were inadvertently interchanged in the plots on the diagonal (pink background) comparing OTU dissimilarity and metagenomic dissimilarity within the same size fraction in Figure 1—figure supplement 2.

However, exchanging the axis labels cannot affect the correlation values that we calculated. Within each size fraction, these OTU dissimilarity and metagenomic dissimilarity Spearman rank-based correlation values ranged from ρ = 0.52 to ρ = 0.98, depending on size fraction (all six correlations were significant at p ≤ 10^-4^). We thus dispute the reviewer’s assertion that “there is no clear correlation between OTUs and metagenomic dissimilarity.”

Among different size fractions, we also obtained high and significant correlation values, both within the metagenomic data and within the OTU data. We would not necessarily expect to find high levels of correlation in community dissimilarity among organisms of such different sizes, life histories, and ecological roles. However, the fact that we do find such correlations reinforces the hypothesis that we propose to explain plankton biogeography, as these organisms are all transported by ocean currents in a similar way. This is confirmed by the correlations we observed (though with lower values) between zooplankton from imagery and all other size fractions, either from metagenomics or OTUs.

The purpose of this figure was to demonstrate the reliability of metagenomic dissimilarity by comparison to previously used measures based on OTUs or on imaging data. We conclude from the results presented in the figure that metagenomic dissimilarity is a reliable measure of community composition differences among samples.

[Editors’ note: what follows is the authors’ response to the second round of review.]

The reviewers and reviewing editor appreciate the extensive work that the authors put into their revised manuscript and the detailed explanations provided in their response letter. There are still some concerns expressed by some of the reviewers, particularly with regards to (a) the ambiguity of using daily sunshine duration as a proxy for seasonality, (b) the potential inclusion of smaller organisms in larger nominal size fractions (for example host-associated microbes may be included in the 180-2000 fraction, thus distorting your analyses), and (c) the focus on surface currents and the omission of the ocean's 3-dimensional structure and variable stratification. The reviewers and reviewing editor have the following minimum recommendations for addressing these issues:1. Please indicate in one of your maps what sampling sites had a mixed water column (e.g. less than 5ºC difference from surface to the subsurface DCM sample) and which ones had a stratified water column (e.g. more than 5ºC difference between surface and subsurface).

We believe that this recommendation was intended to request that we indicate the sampling sites with a greater than 0.5ºC difference from the surface to the DCM, not 5ºC, since typical values in the literature for determining the depth of the mixed layer range from 0.2-1.0ºC (Table 1 in Kara *et al.*, 2000 DOI: 10.1029/2000JC900072). We modified Figure 1—figure supplement 1b (and the associated figure caption) to indicate which stations had more than 0.5ºC difference between surface and DCM samples. There were 42 such stations for which metagenomic data was produced.

For completeness, we also checked for stations with more than a 5ºC difference. There were 9 such stations for which metagenomic data was produced: stations 7, 9, 96, 97, 102, 132, 133, 137, 138.

Temperature recordings for all samples are found in Supplementary Table 2 (unchanged since our original submission).

In the process of studying Figure 1—figure supplement 1b, we noticed that *Tara* Oceans stations 11, 14, 24, 26, 31, 33, 47, 48, 56, 57, 62, 65, 94, 99, 113, 114, 115, 116, 117, 118, 133, 140 and 141 had been inadvertently left out of the map, because an earlier version of this figure applied different criteria for including stations on the map, and we did not update the figure when the criteria were changed. We corrected this error in the modified figure. The full list of surface metagenomic samples is available in Supplementary Table 1 (unchanged since our original submission).

2. Please acknowledge in your paper (e.g., the introduction and discussion) the potential significance of depth, in particular highlighting the point that in the mesopelagic the relationship between composition of plankton communities and currents may be quite different than at the surface.

We added a sentence to the introduction and to the discussion, as follows:

Introduction, lines 135-138: “Our analyses focus on the sunlit (epipelagic) layer of the ocean (subsurface and deep chlorophyll maximum (DCM) samples); at lower depths (the mesopelagic and below), the relationship between plankton community composition and ocean transport may be different than at the surface.”

Discussion, lines 371-373: “At greater ocean depths (i.e., in the mesopelagic and below), the relationship between ocean transport and plankton community composition may differ from the one we describe at the surface.”

3. Please acknowledge (e.g. in the introduction or discussion) that the ocean is a tridimensional system in which the main axis of variation is depth, and that a focus on surface currents is a limitation of this study.

We added two sentences to the discussion, as follows (lines 369-371): “The ocean is a three-dimensional system in which the primary axis of variation is depth. Our metagenomic data and simulations were limited to the sunlit layer of the ocean and therefore capture one part of seascape dynamics.”

4. Please acknowledge in your paper the caveat that daily sunshine duration does not unambiguously map to seasonal effects (since in Spring and Fall daily sunshine durations coincide), and that ocean biology, chemistry and stratification often differ between Fall and Spring.

We added the following, lines 536-541: “…sunshine duration (day length) does not map unambiguously to season, as day lengths coincide in spring and autumn. Ocean biology, chemistry and stratification often differ between spring and autumn. As such, we provide seasonality indices for temperature and for nitrate + nitrite (described above in the Methods, “Sampling, sequencing and environmental parameters”), which represent annual variation in these environmental parameters, and can help interpret the effects of seasonality on genomic provinces.”

We note that these seasonality indices were present in the originally submitted version of our paper (and unchanged in the revised versions), but we did not highlight them when writing our response to the reviewers. We apologize for any confusion this may have caused.

5. Please acknowledge in your paper that your size fractions are operational, i.e., not necessarily mapping precisely to organism sizes but instead a priori only mapping to "whatever is captured between two specific filter pore sizes". Please also provide some supporting information regarding the fraction of microbial (and perhaps even viral reads) present in the larger nominal size fractions, so that the readers can judge to what extent this may have been an issue.

The principal reason for which we chose our approach to calculate plankton β diversity based on comparison of random metagenomic DNA sequences is that it does not depend on knowing the taxonomy nor the genome sequences of the organisms in each community. This feature of our analysis is advantageous both because we lack reference genomes for most planktonic species, and because we readily acknowledge that our size fractions were operational. Due to the latter, we were careful in the text to refer to each size fraction as “enriched” with a certain type of organism, and we refer to organismal size ranges when discussing size fractions, rather than to the taxonomic identity of the community they contain. We note that none of the principal results of the paper (i.e., those presented in Figures 1-3) depends on the taxonomy of the organisms within any given size fraction, although we do refer to particular taxonomic groups (or their biological features) in the discussion, when interpreting our results in the context of the hypothesis we propose for the existence of stable genomic provinces in ocean plankton communities.

We added a sentence to the text to indicate that our size fractions are operational, accompanied by an estimate of the percentages of eukaryotic reads present in the bacterial-enriched size fraction, and the percentages of bacterial reads present in the four eukaryotic-enriched size fractions, lines 144-153, as follows: “Each plankton community sample was sequenced for up to six operational size fractions: one virus-enriched (0-0.22 μm) (Roux et al., 2016), one prokaryote-enriched (either 0.22-1.6 or 0.22-3 μm) (Sunagawa et al., 2015), and four eukaryote-enriched (0.8-5 μm, 5-20 μm, 20-180 μm and 180-2000 μm (de Vargas et al., 2015); Figure 1—figure supplement 1b). These size fractions are operational in that each contains the organisms captured between two physical filters of a given size (either filters or nets, depending on size fraction (Pesant et al., 2015)). We estimated the average percentage of metagenomic sequence reads in samples from the prokaryote-enriched 0.22-1.6/3 µm size fraction that were of eukaryotic origin to be 12%, and the average percentage of reads in eukaryote-enriched size fractions that were of prokaryote origin as follows: 0.8-5 μm: 39%, 5-20 μm: 22%, 20-180 μm: 3%, 180-2000 μm: 5% (see Methods).”

We added a corresponding new section to the Methods, to provide further details on how our estimates were calculated, lines 648-657.

6. Please clarify in line 143 why only 18S sequences are mentioned and not 16S and correct if necessary.

We used miTAGs (metagenomically-derived 16S sequences) to represent bacterial OTUs in our analyses. We modified this line of the text, which now reads (lines 156-161): “We also analyzed Operational Taxonomic Units (OTUs, representing groups of genetically related organisms), consisting of previously published viral populations (Brum et al., 2015) previously derived bacterial 16S miTAGs (Sunagawa et al., 2015), and 738 million 18S V9 ribosomal DNA marker sequences in the eukaryote-enriched size fractions, enlarging a previously described Tara Oceans data set (de Vargas et al., 2015).”

Further details can be found in the Methods (lines 378-384 of the original submission, unchanged in lines 444-450 of this revision): “Within the 0-0.2 μm size fraction, we used previously published viral populations (equivalent to OTUs) (Brum et al., 2015) and viral clusters (analogous to higher taxonomic levels) (Roux et al., 2016) based on clustering of protein content. For the 0.22-1.6/3 μm size fraction, we used previously derived miTAGs based on metagenomic matches to 16S ribosomal DNA loci and processed them as described (Sunagawa et al., 2015). For the four eukaryotic size fractions, we added additional samples to a previously published Tara Oceans metabarcoding data set and processed them using the same methods (de Vargas et al., 2015) (also described at DOI: 10.5281/zenodo.15600).”

7. Please also check in line 142 if 24.2 TB of data were indeed analyzed, or if this is the total number of sequences but not all were actually analyzed.

We modified this line of the text, which now reads (lines 153-156): “The *Tara* Oceans project produced a total of 24.2 terabases of metagenomic sequence reads (Supplementary Table 1). To account for uneven sequencing depth among samples, we analyzed a subset of 11.9 terabases, after testing that this subset accurately represented the complete data set (see Methods).”